# Clinical Benefits of Exogenous Ketosis in Adults with Disease: A Systematic Review

**DOI:** 10.3390/nu17193125

**Published:** 2025-09-30

**Authors:** Othmane Mohib, Sarah Bomans, Berenice Jimenez Garcia, Lynn Leemans, Claudine Ligneel, Elisabeth De Waele, David Beckwée, Peter Janssens

**Affiliations:** 1Department of Nephrology and Arterial Hypertension, Vrije Universiteit Brussel (VUB), Universitair Ziekenhuis Brussel (UZ Brussel), Laarbeeklaan 101, 1090 Brussels, Belgium; peter.janssens@uzbrussel.be; 2Vitality Research Group, Vrije Universiteit Brussel (VUB), 1090 Brussels, Belgium; lynn.leemans@uzbrussel.be (L.L.); elisabeth.dewaele@uzbrussel.be (E.D.W.); 3Department of Clinical Nutrition and Dietetics, Vrije Universiteit Brussel (VUB), Universitair Ziekenhuis Brussel (UZ Brussel), Laarbeeklaan 101, 1090 Brussels, Belgium; sarah.bomans@uzbrussel.be (S.B.); berenice.jimenezgarcia@uzbrussel.be (B.J.G.); 4Department of Clinical Pharmacology and Pharmacotherapy, Universitair Ziekenhuis Brussel (UZ Brussel), Laarbeeklaan 101, 1090 Brussels, Belgium; claudine.ligneel@uzbrussel.be; 5Rehabilitation Research Department, Vrije Universiteit Brussel (VUB), 1090 Brussels, Belgium; david.beckwee@vub.be

**Keywords:** exogenous ketosis, ketone ester, medium-chain triglycerides, ketone salt, ketosis

## Abstract

Background/Objectives: Ketone bodies are increasingly studied for their potential therapeutic effects, particularly through exogenous ketosis, in a variety of diseases. This systematic review aimed to rigorously assess the clinical efficacy of exogenous ketosis in adults with medical conditions. Methods: Following PRISMA guidelines, we systematically searched MEDLINE and Scopus databases. Our inclusion criteria were defined according to the PICOS framework, focusing on studies involving exogenous ketosis in adult patients with specific diseases. The study is registered in the International Prospective Register of Systematic Reviews (PROSPERO; CRD42023492846). Results: After a stringent selection process, fifty-one studies were analyzed. Twenty-two studies focused on neurological disorders, one on psychiatric disorders, twenty-two on metabolic disorders, five on cardiovascular disorders, and one on an inflammatory disorder. Exogenous ketosis demonstrated potential benefits across multiple conditions, including Alzheimer’s disease, mild cognitive impairment, McArdle’s disease, various forms of heart failure, cardiogenic shock, pulmonary hypertension, and COVID-19-related acute respiratory distress syndrome, although evidence is mostly limited to surrogate endpoints with insufficient hard outcome data. Subtherapeutic ketone concentrations induced by medium-chain triglycerides and limited follow-up periods often precluded firm conclusions regarding clinically meaningful outcomes. Conclusions: Exogenous ketosis shows potential in neurological, metabolic, and cardiovascular disorders, while evidence in psychiatric and inflammatory conditions remains scarce and preliminary. Ketone esters appear preferable for effective and tolerable ketosis. Future research should focus on identifying responsive patient populations, optimizing treatment regimens, and conducting long-term clinical trials with hard endpoints to validate these findings.

## 1. Introduction

Ketone bodies are fundamental metabolites that are best known for their function as an alternative fuel source to glucose during energy restriction with reduced carbohydrate intake [1]. In this physiological state, ketogenesis in the liver converts fatty acids into β-hydroxybutyrate (βHB), acetoacetate (AcAc), and acetone. The main characteristics of ketone bodies and the different ways to induce ketosis are described in Figure 1. βHB and AcAc can then serve as an energy source for extrahepatic tissues. βHB is the most abundant and stable ketone body [1]. In a normal fed state, circulating βHB levels are typically less than 0.5 mmol/L but this can rise to 6–7.5 mmol/L during extended fasting [2]. Ketosis is defined as a blood βHB level greater than 0.5 mmol/L [2].

Ketosis can be achieved endogenously or exogenously. Endogenous ketosis occurs when the liver produces ketone bodies from free fatty acids, typically triggered by a ketogenic diet (low carbohydrate, moderate protein, high fat) or intermittent fasting. Both methods can effectively induce ketosis [3,4]. However, depending on its composition and the individual’s metabolic profile, a ketogenic diet may be associated with elevations in low-density lipoprotein cholesterol (LDL-c) and Apolipoprotein B [5]-biomarkers linked with atherosclerosis risk- and can also lead to nutritional deficiencies [6] or nephrolithiasis [7]. Moreover, compliance rates with these regimens are poor, reaching only 38% to 56% in some cases [8], with systematic reviews showing short-term adherence around 66–80% but declining to about 38% after three years [9]. Adherence to intermittent fasting is also poor, with a drop-out rate of 38% in a major study [10].

Exogenous ketosis can be achieved alternatively through the use of exogenous ketone supplements. Three main categories of exogenous ketone supplements are available: ketone salts, ketone esters, and medium-chain triglyceride (MCT). Ketone salts are composed of βHB bound to a mineral (potassium, sodium, or magnesium) [11,12]. Ketone esters often consist of a ketone body bound to a precursor molecule such as 1,3-butanediol, that is typically also metabolized to a ketone body. Ketone esters are more effective in raising blood ketone levels and induce fewer adverse events compared to ketone salts [11,13]. MCTs are highly ketogenic fats that are metabolized to ketone bodies in the liver with direct intestine-to-liver portal absorption [14]. MCTs are not exogenous ketone supplements per se, they act as exogenous ketogenic substrates and are pragmatically regarded as exogenous ketogenic supplements in clinical practice. More recently, additional exogenous ketones have emerged, such as pure ketone diol (containing exclusively 1,3-butanediol) or free bioidentical βHB [15]. While daily administration of diluted free βHB (10 g) once a day for 28 days in healthy participants was well tolerated and safe [15], these molecules have accumulated less clinical experience.

In therapeutic studies using exogenous ketosis, βHB concentrations typically range between 1.0 and 3.0 mmol/L with ketone esters, and often remain below 1.0 mmol/L when using MCTs [11,12]. Reported adverse events associated with the use of exogenous ketone supplements are generally mild and include gastrointestinal symptoms such as nausea, bloating, and diarrhea, particularly with higher doses or with MCT-based approaches [14].

**Figure 1 nutrients-17-03125-f001:**
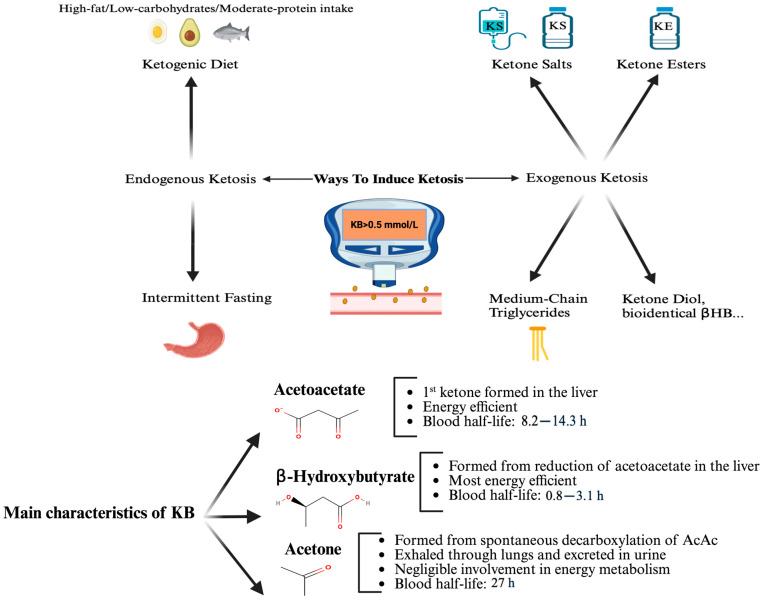
The main characteristics of ketone bodies and the different ways to induce ketosis. AcAc, acetoacetate; KB, ketone bodies; βHB, β-hydroxybutyrate; KS, ketone salt; KE, ketone ester. Adapted from [16,17].

Beyond serving as an alternative energy source, ketone bodies also serve as signaling molecules. βHB has been shown to act as an effector via G-protein coupled receptors, such as the hydrocarboxylic acid receptor 2 [18], modulating cellular homeostasis of many cell subtypes. By inhibiting the pro-inflammatory transcription factor nuclear factor-kappa B (NF-κB) and enhancing autophagy, βHB can provide anti-inflammatory effects, leading to decreased levels of proinflammatory mediators like interleukin-1β (IL-1β) and tumor necrosis factor α (TNF-α)) [19]. Furthermore, βHB has been shown to decrease oxidative stress by increasing the expression of antioxidant genes [20] and improving mitochondrial function [21]. βHB also functions as an epigenetic gene regulator by inhibiting class I histone deacetylases. This leads to an increase in histone acetylation, influencing the expression of genes that reduce oxidative stress and enhance mitochondrial homeostasis [22]. Finally, βHB can influence appetite and satiety [23], inhibit sympathetic nervous system activity, and reduce heart rate (HR) and total energy expenditure [24].

These pleiotropic effects hold therapeutic potential in several diseases, explaining an increased interest in ketosis among researchers, clinicians, and the general public. Initially, interest in ketosis—induced by a ketogenic diet or intermittent fasting—emerged several decades ago as a treatment for epilepsy and diabetes mellitus [25,26]. More recently, the widespread availability of exogenous ketone supplements has spurred claims of diverse health benefits, underscoring the need for a thorough evaluation of the scientific evidence.

Although several systematic reviews have examined exogenous ketosis, most have either focused on healthy populations or restricted their scope to a single disease domain [27,28]. To our knowledge, this is the first systematic review to focus exclusively on adults with established medical conditions, deliberately excluding healthy individuals. By synthesizing evidence from various disorders, our review provides a comprehensive, clinically oriented overview. With rigorous methodology, PROSPERO registration, and an updated search through February 2025, our review not only assesses therapeutic potential but also the methodological limitations and knowledge gaps for future research.

## 2. Methods

A protocol was drafted for this systematic review in accordance with the Preferred Reporting Items for Systematic Review and Meta-Analysis Protocols (PRISMA-P) checklist [29]. This systematic review was registered in the International Prospective Register of Systematic Reviews (PROSPERO) on 12 December 2023, with the following number: CRD42023492846. The study was conducted according to the PRISMA statement [30].

### 2.1. Search Strategy

The MEDLINE and Scopus databases were explored. Although the original protocol specified four databases, including the Cochrane Database of Systematic Reviews and PROSPERO, these databases were not searched because systematic reviews were listed as an exclusion criterion. The search terms and key concepts used were as follows: (Exogenous ketogenic supplements) OR (exogenous ketones supplements) OR (ketone supplementation) OR (ketone supplements). Articles published from the inception of each database until 5 December 2023 were retrieved. We subsequently updated our systematic review with a new database search on 13 February 2025. There were no geographical restrictions but languages other than English were excluded.

In addition, the reference lists of retrieved papers were screened to identify other articles of interest. The grey literature was not searched.

### 2.2. Inclusion and Exclusion Criteria

The inclusion and exclusion criteria applied in this systematic review (Table 1) were proposed according to the Participants, Interventions, Comparators, Outcomes, and Study design (PICOS) framework.

Studies were eligible if they assessed at least one clinical, biological, or radiological outcome, or adverse event, regardless of whether the results demonstrated improvement, harm, or no effect.

### 2.3. Selection Process

Two reviewers (OM, SB) conducted independent searches for relevant studies and determined which ones should be included (using Rayyan). The records were managed using Rayyan, a free web tool (Rayyan.ai). Rayyan detects duplicates by considering various factors such as the title, authors, journal, and year. A second check for duplicate articles was carried out by two reviewers (OM, SB).

After removing duplicates, titles and abstracts were screened. The full text of a study was reviewed, and the study was considered to be potentially relevant when it could not clearly be excluded based on its title and abstract following discussion between the two independent reviewers. In the event of disagreement or inadequate information in an abstract, the full text was sought. When a study was determined by both reviewers to meet the inclusion criteria from the complete text, it was included. After discussion, if there was a disagreement, a third reviewer (PJ) mediated.

### 2.4. Data Extraction, Risk of Bias Assessment and Analysis

Two reviewers extracted the data independently via a standardized extraction form in Excel including the following summary data: reference (year), condition, study design, population (characteristics and size), type and dose of exogenous ketone supplement, ketone blood levels, primary outcome measures, other outcome measures, results, and study limitations.

Risk of bias was assessed using the Risk Of Bias In Non-randomized Studies-of Interventions (ROBINS-I) in non-randomized controlled trials and the Cochrane Risk of Bias tool in randomized controlled trials [31]. The same initial reviewers (OM, SB) independently evaluated the risk of bias for each trial included.

Due to the heterogeneity of study populations, interventions, and outcomes, a quantitative synthesis was not feasible. Instead, we conducted a structured narrative synthesis. Studies were grouped by disease domain (neurological, psychiatric, metabolic, cardiovascular, inflammatory), and findings were tabulated to allow comparison of study characteristics, type and dose of exogenous ketone, achieved βHB levels, and primary outcomes. Within each disease domain, results were summarized narratively to highlight consistencies, divergences, and limitations.

## 3. Results

### 3.1. Study Selection and Characteristics

The initial literature search retrieved 3941 articles. After removing duplicates, 3085 records remained. Next, a screening of the titles and abstracts was conducted. A total of 3031 records were excluded and 54 articles remained. The full texts of the remaining 54 articles were then carefully reviewed and assessed for eligibility. Finally, 51 articles [32,33,34,35,36,37,38,39,40,41,42,43,44,45,46,47,48,49,50,51,52,53,54,55,56,57,58,59,60,61,62,63,64,65,66,67,68,69,70,71,72,73,74,75,76,77,78,79,80,81,82] were chosen that met the inclusion criteria. A PRISMA statement flow diagram illustrating the study selection process is shown in Figure 2.

The distribution of included studies by disorder type is detailed in Table 2.

Among these, 51 included articles and half (25/51) involved ketone esters [49,55,56,57,58,59,60,61,62,63,64,65,66,68,69,70,71,73,74,75,76,77,79,80,82]. In 21 studies [32,33,34,35,36,37,38,39,40,41,42,43,44,45,46,47,48,50,51,53,72], MCTs were used and, in the remaining 5 studies [52,54,67,78,81], ketone salts were administered.

The main characteristics of studies on neurological, psychiatric, metabolic, cardiovascular, and inflammatory disorders are presented in Table 3, Table 4, Table 5, Table 6, and Table 7, respectively. Within these tables, it has been decided to report blood βHB levels expressed as mean (+/− standard deviation), with some exceptions which will be mentioned. The limitations reported correspond to those mentioned by the authors of each article.

### 3.2. Risk of Bias in Studies

The risk of bias graph for randomized controlled trials, using the Cochrane Risk of Bias tool, is available as Appendix A, and the results of the ROBINS-I (traffic light plot) used to assess the risk of bias in non-randomized controlled trials or uncontrolled trials are available as Appendix A.

#### 3.2.1. Randomized Controlled Trials

Most RCTs were judged at low-to-moderate risk of bias. Randomization and allocation concealment were often insufficiently described, and blinding was unclear in several trials, particularly older MCT studies. Attrition bias due to high dropout rates (notably in Alzheimer’s trials using MCTs) was common.

#### 3.2.2. Non-Randomized and Open Label Studies

Across five non-randomized studies [35,36,53,63,77], overall risk of bias was mostly moderate, but confounding was serious in four studies [35,53,63,77], reflecting the absence of comparators and lack of adjustment. Selection bias was serious in two small, highly selected cohorts [36,53]. In contrast, classification of interventions was uniformly low risk, and deviations, missing data, measurement, and reporting ranged from low to moderate, with the lowest risks in studies using objective physiological outcomes [36,77].

#### 3.2.3. Systematic Patterns

Blinding and allocation concealment were the most frequent unclear domains in RCTs. Confounding was the most problematic domain in non-RCTs.

### 3.3. Results of Individual Studies

#### 3.3.1. Neurological Disorders

Twenty-two studies [32,33,34,35,36,37,38,39,40,41,42,43,44,45,46,47,48,49,50,51,52,53] investigated neurological disorders, focusing primarily on Alzheimer’s disease (AD) and mild cognitive impairment (MCI). Across these studies, supplementation with MCTs was generally associated with modest improvements in cognitive domains such as memory, executive function, and language, particularly among Apoε4-negative individuals. However, the extent of ketosis achieved was typically low (<0.6 mmol/L), which limited the magnitude of clinical benefit. Dropout rates were high and biomarker confirmation was lacking, thereby reducing confidence in the findings. Other neurological conditions showed more heterogeneous outcomes.

For multiple sclerosis (MS), two trials [50,51] reported reductions in anxiety, depressive symptoms, and abdominal fat distribution, but interpretation was hampered by concurrent dietary interventions.

A trial in migraine patients [52] using ketone salts failed to demonstrate reductions in migraine frequency or severity, most likely because of insufficient ketosis.

In Parkinson’s disease [49], a small exploratory study with ketone esters demonstrated improved endurance performance.

An uncontrolled epilepsy trial [53] suggested seizure reduction with MCTs but was at very high risk of bias.

Overall, neurological evidence is promising but inconsistent, and methodological limitations prevent firm conclusions.

#### 3.3.2. Psychiatric Disorders

In psychiatric disorders, evidence is currently limited to a single small randomized controlled trial in post-traumatic stress disorder (PTSD) [54]. This study suggested potential symptom improvement with ketone salt supplementation, but no between-group differences were detected, and blood ketone levels were not measured. Thus, evidence in psychiatric conditions remains very preliminary.

#### 3.3.3. Metabolic Disorders

Metabolic disorders were investigated in twenty-two studies [55,56,57,58,59,60,61,62,63,64,65,66,67,68,69,70,71,72,73,74,75,76].

Acute ketone ester supplementation in pre-diabetes and type 2 diabetes consistently improved glycemia (only in fasting conditions in type 2 diabetes), insulin response, and lipid parameters. Longer-term exogenous ketosis (2–4 weeks) showed a beneficial effect on HbA1c [63], but conflicting results on fructosamine [63,64], though limited by small sample sizes.

In obesity, ketone esters reduced post-prandial glucose, improved vascular function, and enhanced some aspects of cognitive performance.

Rare metabolic diseases such as very-long-chain acyl-CoA dehydrogenase deficiency (VLCADD) and McArdle disease were also studied. In VLCADD [74], ketone ester supplementation improved muscle bioenergetics, while in McArdle disease, results were inconsistent: one ketone ester trial [75] showed no improvement in exercise tolerance, while another [76] reported worsened performance compared to carbohydrate supplementation.

#### 3.3.4. Cardiovascular Disorders

Five studies addressed cardiovascular disorders [77,78,79,80,81].

In heart failure with reduced ejection fraction (HFrEF) [77,78], both ketone esters and salts acutely improved cardiac output (CO), stroke volume, and ventricular function, with consistent reductions in systemic vascular resistance (SVR).

In heart failure with preserved ejection fraction (HFpEF) [79], short-term ketone ester supplementation increased cardiac output, improved diastolic function, and reduced pulmonary capillary wedge pressure both at rest and during exercise, although potential carry-over effects could not be excluded.

In cardiogenic shock [80], ketone esters improved CO, cardiac power output, and systemic oxygen delivery without adverse safety signals.

In pulmonary hypertension [81], ketone salts increased CO and right ventricular contractility, though effects were largely acute and may have been influenced by hemodynamic changes induced by sodium load.

Overall, cardiovascular studies consistently demonstrated improvements in hemodynamic parameters and myocardial energetics, though evidence remains limited to short-term interventions in small samples.

#### 3.3.5. Inflammatory Disorders

Finally, evidence in inflammatory disorders derives from a single randomized controlled trial in patients with COVID-19-related acute respiratory distress syndrome [82]. In this study, supplementation with a βHB formulation reduced pro-inflammatory cytokines, increased anti-inflammatory mediators, improved oxygenation, and shortened hospital stay. These results suggest potential immunomodulatory effects, but the evidence remains preliminary and requires replication in larger, longer-term studies.

### 3.4. Synthesis Across Studies

Across the diverse disease domains studied, several broad patterns can be discerned. The most consistent benefits of exogenous ketosis were observed in metabolic regulation and cardiovascular performance, where ketone esters in particular improved glycemic control, lipid metabolism, and hemodynamic function.

Neurological outcomes were more variable: while some improvements in cognition were reported in AD and MCI, especially in Apoε4-negative individuals, effects were generally modest and limited by low levels of ketosis in MCT-based studies.

Non-randomized and open-label studies tended to report more favorable outcomes than randomized trials, underscoring the importance of rigorous methodology. Across all domains, however, clinically meaningful endpoints such as long-term disease progression, quality of life, and survival remain underreported. The evidence base as a whole is characterized by small sample sizes, heterogeneous designs, and reliance on surrogate outcomes. Taken together, the current findings indicate promising therapeutic potential of exogenous ketosis, but confirmatory large-scale trials with long-term follow-up are needed to establish clinical efficacy and safety across different patient populations.

## 4. Discussion

### 4.1. Summary of Main Findings

This systematic review evaluated the clinical impact of exogenous ketosis in various diseases. Fifty-one clinical studies were included, mostly focusing on neurological, metabolic, and cardiovascular disorders. Exogenous ketosis demonstrated potential benefits on cognitive function, metabolic regulation, cardiac performance, and inflammatory processes. However, therapeutic responses were heterogenous and depended on factors such as ketone type, dosage, and patient characteristics. The majority of evidence is based on surrogate markers rather than hard clinical outcomes, and the duration of follow-up was typically limited.

#### 4.1.1. Neurological Disorders

*Therapeutic insights and mechanisms of action:* Exogenous ketosis shows particular promise in AD and MCI, conditions associated with impaired cerebral glucose metabolism [83,84]. Ketone bodies may partially compensate by serving as alternative cerebral energy substrates, while also exerting anti-inflammatory and antioxidant properties via inhibition of NF-kB signaling and enhancing mitochondrial function [85,86,87,88].

In the studies reviewed, ketone supplementation was associated with modest improvements in variable outcome measures for episodic memory, executive function, and language skills, particularly in Apoε4-negative individuals who appear to respond more favorably to ketone supplementation. Six-month administration of MCTs had some effects on various circulating cardiometabolic and inflammatory markers.

In the context of MS [50,51], exogenous ketosis may reduce inflammation, anxiety, depression, and abdominal fat in MS patients. The benefits appear linked to ketone body–mediated neuroprotection, gamma-aminobutyric acid (GABA)-ergic [89] and N-methyl-D-aspartate modulation [90], and improved metabolic regulation.

Ketone bodies are considered [91] to stabilize brain excitability by enhancing GABAergic tone, reducing glutamatergic activity, and limiting cortical spreading depression, a key process in migraines. Ketones may also improve mitochondrial efficiency and reduce oxidative stress, potentially lowering susceptibility to migraine triggers. However, the relatively low systemic ketone levels achieved [52] suggest that therapeutic benefit depends on attaining higher and more sustained ketosis.

Finally, MCTs support ketosis and generate decanoic acid, which exerts direct antiseizure effects through inhibition of excitatory AMPA receptors [92]. Beyond seizure control, these mechanisms highlight the broader potential of MCT-induced ketosis for stabilizing neuronal networks. By combining ketone body–mediated neuroprotection, anti-excitatory receptor modulation, and improved cerebral energy metabolism, MCTs may represent a targeted metabolic strategy relevant to epilepsy.

Taken together, these findings suggest that the effects of ketone bodies in neurological disorders are not limited to their role as an alternative fuel source. Rather, they appear to act through multiple converging mechanisms, including improved mitochondrial bioenergetics, reduction of oxidative stress, modulation of excitatory–inhibitory neurotransmission, and attenuation of neuroinflammation. Importantly, these mechanisms are shared across distinct disorders such as AD, MS, migraine, and epilepsy, suggesting that exogenous ketosis may represent a unifying metabolic strategy for stabilizing neuronal function and resilience.

*Limitations in existing studies and future directions:* Clinical results are modest at best. Most clinical trials utilized MCTs, achieving relatively low βHB levels (often <0.5 mmol/L), which may be insufficient to elicit robust clinical results. Additionally, studies with MCTs often faced high dropout rates (up to 25% of participants in the interventional group in one trial [33]) due to gastrointestinal side effects, potentially compromising adherence and outcome reliability. A gradual increase in MCT dose [35] could improve tolerance and compliance.

A notable finding in MCI and AD contexts is the differential response based on Apoε4 status [32,33,37,44], with Apoε4-negative participants often consistently showing greater cognitive benefits, indicating the need for genetic stratification in future studies.

More broadly, when assessing the relevance of exogenous ketosis for a particular neurological disease, inter-individual variability in disease severity needs to be considered. Many disorders (e.g., migraine, AD) are heterogeneous conditions, potentially limiting efficacy to subgroups of patients.

Further research is warranted to explore the long-term efficacy of ketone esters in neurological disorders, as these formulations induce potent sustained ketosis with fewer side effects. Given the heterogeneous nature of AD and MCI, future studies should aim to identify biomarkers that predict response to exogenous ketosis, with a focus on distinguishing responders from non-responders based on genetic, metabolic, and inflammatory profiles.

#### 4.1.2. Psychiatric Disorders

*Therapeutic Insights and Mechanisms of Action:* Exogenous ketosis in PTSD subjects [54] may enhance standard pharmacological treatment by modulating neurotransmitter systems and reducing neuroinflammation [93]. These mechanisms suggest potential benefits in symptom reduction, faster response, and improved overall outcomes.

*Limitations in Existing Studies and Future Directions:* As a pilot trial [54], this study is limited by small sample size, short duration, and heterogeneity among participants, restricting generalizability. Future research should include larger, longer trials with biomarker assessments to clarify mechanisms, identify responsive subgroups, and establish long-term safety and efficacy.

#### 4.1.3. Metabolic Disorders

*Therapeutic insights and mechanisms of action:* Exogenous ketosis may be associated with improved glycemic control, insulin sensitivity, and lipid metabolism, particularly in pre-diabetic and obese individuals. Ketone esters and MCT lowered blood glucose and triglycerides while modestly increasing insulin levels. However, studies were performed in selected populations and results are not uniform. Ketone bodies may modulate glucose metabolism through multiple pathways, including increased insulin secretion [94], suppression of lipolysis [18], and/or a reduction of gluconeogenesis [95].

One cohort [55,56,57,58,59,60,61] evaluated the acute impact of exogenous ketosis in a specific cohort of new-onset prediabetes after pancreatitis. As in healthy and obese populations [68,96], acute exogenous ketosis may lead to a decrease in blood glucose levels, modulated by abdominal fat distribution [56].

In type II diabetic participants, both acute and chronic administration of ketone ester in one study [64] did not lead to a detectable drop in blood fasting glucose or fructosamine levels, but significantly and slightly increased insulin without a significant variation in C-peptide. It was hypothesized that the increase in circulating insulin may not have been sufficient to affect glucose clearance or hepatic gluconeogenesis, in the context of insulin resistance. Also, a decreased insulin-secreting capacity of β cells alongside insulin resistance minimize any insulin-mediated reductions in blood glucose. However, in contrast with this clinical trial [64], in which a single dose of ketone ester was administered under fasting conditions and away from a meal, Monteyne et al. [66] proved that ketone ester administration (vs. placebo) with a liquid mixed-meal tolerance test using a dual-glucose tracer approach leads to a significant reduction in post-prandial glucose concentrations, mainly as a result of a decrease in the 2 h rate of glucose appearance following meal ingestion. As no effect on post-prandial endogenous glucose production nor plasma insulin were demonstrated, the authors concluded that reduced post-prandial glucose concentration within this type II diabetes population may be induced by a delayed glucose absorption. A prior systematic review with meta-analysis had already demonstrated the blood glucose–lowering effect of exogenous ketosis. However, the majority of included studies were conducted in healthy participants, with only 6 of the 51 studies involving individuals with type II diabetes, prediabetes, or obesity [96].

Next, while βHB demonstrated anti-inflammatory effects via the NLRP3 inflammasome pathway in preclinical studies [97,98], exogenous ketosis did not inhibit NLRP3 inflammasome activation in a clinical study [69], likely due to an exposition of monocytes to weak levels of βHB and LPS.

Finally, since obesity is also associated with neurocognitive dysfunction [99], the effects of ketone ester supplementation were studied and early data suggest potential cognitive benefits with an improvement in cerebral blood flow [70,73].

Bleeker et al. [74] induced exogenous ketosis prior to exercise in patients with VLCADD. The uptake of βHB by skeletal muscles was significant and blood glucose levels remained normal, suggesting utilization of ketone bodies as an energy substrate. A lower Pi/PCr ratio in leg muscle during exercise suggested an improved intramuscular energy balance.

Patients with McArdle disease have blocked glycogen breakdown and ketone bodies could serve as an alternative energy fuel. However, exercise capacity was not significantly improved after induction of exogenous ketosis [75]. This was explained by the fact that ketone bodies induce inhibition of lipolysis [11] and liver glucose output [100], leading to reduced free fatty acid (FFA) and glucose availability in the muscle. A recent study [76] confirmed these results, demonstrating an important impairment in exercise capacity following acute ketone ester supplementation in McArdle disease.

*Limitations in Existing Studies and Future Directions*: Most studies in metabolic disorders assessed only the acute effects of ketosis and often lacked long-term follow-up, limiting the conclusions on sustained benefits. Additionally, a variability in ketone sources, dosages, and the metabolic profiles of participants introduces significant heterogeneity. The induction of exogenous ketosis might be of benefit in various diseases resulting from inborn errors of metabolism, including multiple acyl-CoA dehydrogenation deficiency. Future trials should evaluate the long-term effects of ketone esters in larger, well-characterized cohorts, focusing on outcomes such as insulin sensitivity, body composition, and cardiovascular risk markers.

#### 4.1.4. Cardiovascular Disorders

*Therapeutic Insights and Mechanisms of Action:* In the setting of HFrEF, exogenous ketosis was associated with improved CO and left ventricular function, by increasing myocardial efficiency and lowering SVR [78,101]. βHB may enhance cardiac energy efficiency by serving as an oxygen-sparing fuel source, thus improving mechano-energetic coupling in heart failure. Ketone bodies may also act as agonists for cellular receptors involved in HR regulation [102], explaining the observed increase in HR. It is very interesting to note that despite the increase in cardiac work, βHB did not deteriorate mechano-energetic coupling in terms of myocardial external efficiency (MEE), as βHB similarly increased cardiac work and MVO2. Given that a decrease in MEE is associated with a worse prognosis [103], the maintenance of an invariable MEE is a strong argument for the administration of βHB in these patients. Moreover, myocardial βHB fractional extraction increased linearly in proportion to ketone delivery with no upper threshold and correlated positively with hallmarks of LV remodeling [77]. Finally, the increased βHB fractional extraction was associated with an enhanced uptake of lactate, without any variation of extraction of glucose or FFA, indicating improved myocardial metabolic flexibility.

Even in the situation of cardiogenic shock and ongoing vasoactive medication, enteral treatment with ketone ester may be associated with an improved CO and an increase in mixed venous oxygen saturation (SVO2), without any significant variation of mean arterial pressure [80]. Moreover, as suggested by the lower blood glucose levels and circulating FFAs during ketone ester supplementation while insulin levels remained similar between treatments, ketone ester administration may help to minimize the vicious metabolic cycle of cardiogenic shock by enhancing insulin sensitivity and limiting excessive stress-induced lipolysis.

In HFpEF, ketone esters were associated with an increased CO and also reduced cardiac filling pressures and myocardial stiffness at rest and during exercise [79]. Since inflammation has been shown to play a major role in cardiac stiffness in HFpEF [104], the authors hypothesized that the anti-inflammatory activity of ketone bodies would explain the reduction in cardiac stiffness.

Finally, an increase in CO and a decrease in pulmonary vascular resistance (PVR) following the induction of exogenous ketosis were also observed in patients with PH [81]. However, despite this decrease in PVR, a small increase in mean pulmonary arterial pressure (mPAP) was reported. The authors hypothesized that the decrease in PVR was due to the recruitment of pulmonary vessels following the increased in CO [105], as seen during exercise [106].

*Limitations in Existing Studies and Future Directions*: Small sample sizes and short intervention durations hinder the generalizability of findings. The hemodynamic benefits observed may also be contingent on the high βHB levels achieved with ketone esters, underscoring the need for dosage standardization in future research. Larger clinical trials with longer follow-up periods could help establish the role of exogenous ketosis as an adjunct therapy in cardiovascular diseases.

#### 4.1.5. Inflammatory Disorders

*Therapeutic Insights and Mechanisms of Action:* Severe acute respiratory syndrome coronavirus 2 (SARS-CoV-2) can trigger ARDS. Activation of the NF-κB signaling pathway represents a major contribution in SARS-CoV-2-induced cytokine storm [106]. In the setting of mild COVID-19-related ARDS [82], exogenous ketosis may be associated with a decreased inflammation, increased peripheral oxygen saturation, and reduced duration of hospitalization and muscle fatigue.

*Limitations in Existing Studies and Future Directions:* The efficacy of exogenous ketosis in a population with severe COVID-19-related ARDS has yet to be demonstrated. Future studies should assess the impact of exogenous ketosis on other inflammatory processes.

### 4.2. Strengths and Limitations of the Review

This comprehensive overview of clinically relevant outcomes of exogenous ketosis in adults with various diseases offers an accessible summary for clinicians and researchers. The results highlight multiple challenges in interpreting the efficacy of exogenous ketosis across diverse populations. The heterogeneity in study designs, patient populations, types of ketone supplements, and outcome measures, as well as the short duration of most studies, limits the comparability of findings and restricts conclusions on long-term safety and effectiveness on clinically relevent endpoints. Risk of bias was also variable, which further reduces certainty of the evidence. Compared with previous systematic reviews, which often included healthy participants or focused on single disease areas or ketogenic diets, our review uniquely synthesizes evidence on exogenous ketosis in adults with established medical conditions. Taken together, this provides a complementary perspective while underscoring the need for larger, well-controlled, long-term clinical trials. Finally, the database search was restricted to MEDLINE and Scopus; while these databases provide broad coverage and considerable overlap with Embase and Cochrane CENTRAL, some relevant studies may not have been captured.

## 5. Conclusions

This systematic review highlights that exogenous ketosis may hold therapeutic potential across a spectrum of diseases, including neurological, psychiatric, metabolic, cardiovascular, and inflammatory conditions. However, the current body of evidence is insufficient to establish clear clinical efficacy. Most available studies are limited by small sample sizes, short follow-up periods, high dropout rates, reliance on surrogate endpoints, and methodological constraints, which preclude firm conclusions regarding long-term or clinically meaningful benefits. While ketone esters appear more effective than other formulations in achieving sustained ketosis with acceptable tolerability, the overall results remain heterogeneous and modest. Future high-quality, adequately powered randomized controlled trials with longer follow-up and clinically relevant hard outcomes are needed before exogenous ketosis can be considered a robust therapeutic strategy.

## Figures and Tables

**Figure 2 nutrients-17-03125-f002:**
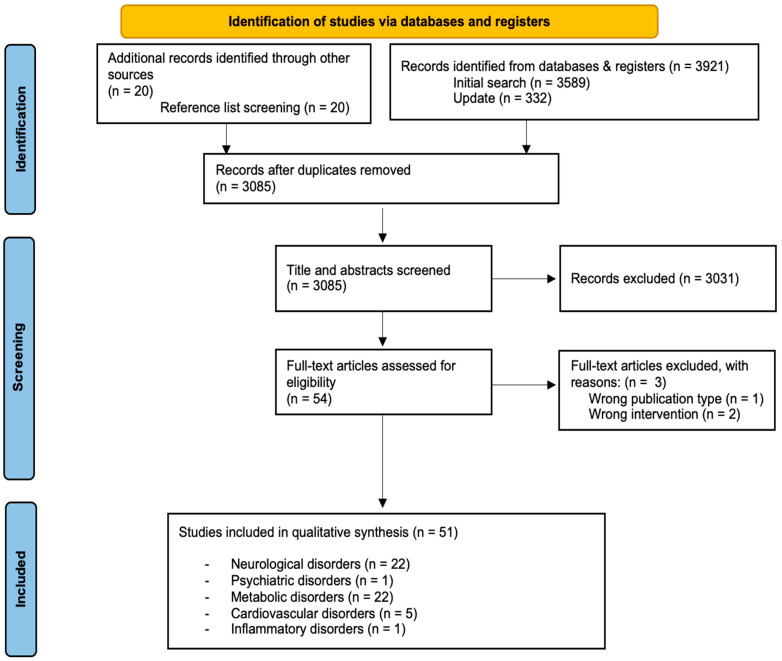
PRISMA flow diagram of the study selection process.

**Table 1 nutrients-17-03125-t001:** Inclusion and exclusion criteria based on PICOS algorithm.

**Inclusion Criteria**	**Patient (P)**	**Adults with Disease**
	Intervention (I)	Induction of exogenous ketosis via: - Ketogenic fats (MCTs) - Synthetic compounds such as ketone salts or ketone esters
	Comparison (C)	- No intervention - Placebo - Standard of care/active medication for the disease
	Outcome (O)	Assessment of disease-related outcomes, including clinical, biological, or radiological parameters, or adverse events
	Study design (S)	- Clinical trials (randomized and non-randomized controlled trials, open-label clinical trials)
**Exclusion Criteria**	Patient (P)	- Children - Adolescents - Adults in healthy condition
	Intervention (I)	Induction of endogenous ketosis via: - Ketogenic diet - Starvation - Diabetic ketosis
	Comparison (C)	/
	Outcome (O)	/
	Study design (S)	Animal studies, in vitro studies, observational studies, reviews, opinion articles, guidelines, letters, editorials, comments, case reports/series, abstracts, dissertations, theses, and articles published in any other language than English

**Table 2 nutrients-17-03125-t002:** Distribution of included studies by disorder type.

Type of Disorder	Disease	N° Studies Included with References
Neurological	MCI	5 [44,45,46,47,48]
	AD	11 [33,34,35,36,37,38,39,40,41,42,43]
	MCI and AD	1 [32]
	PD	1 [49]
	Multiple sclerosis	2 [50,51]
	Episodic migraine	1 [52]
	Drug-resistant epilepsy	1 [53]
Psychiatric	PTSD	1 [54]
Metabolic	Pre-diabetes	8 [55,56,57,58,59,60,61,62]
	Type II diabetes mellitus	5 [63,64,65,66,67]
	Obesity	6 [68,69,70,71,72,73]
	VLCADD	1 [74]
	McArdle disease	2 [75,76]
Cardiovascular	HFrEF	2 [77,78]
	HFpEF	1 [79]
	Cardiogenic shock	1 [80]
	PH	1 [81]
Inflammatory	COVID-19-related ARDS	1 [82]

Abbreviations: AD: Alzheimer’s disease; ARDS: acute respiratory distress syndrome; COVID-19: coronavirus disease 2019; HFpEF: heart failure with preserved ejection fraction; HFrEF: heart failure with reduced ejection fraction; MCI: mild cognitive impairment; PD: Parkinson’s disease; PH: pulmonary hypertension; PTSD: post-traumatic stress disorder; VLCADD: very long-chain acyl-CoA dehydrogenase deficiency.

**Table 3 nutrients-17-03125-t003:** Overview of studies on neurological disorders.

Condition	Reference (Year)	Study Design	Participants (N)	Type of Exogenous Ketone	Dose of Exogenous Ketone	Blood βHB Level	Primary Outcomes	Main Findings	Main Limitations
**MCI**	Myette-Côté et al. (2021) [48]	Double-blinded RCT (vs. placebo)	39 (intervention: 19; placebo: 20)	MCT (60% caprylic + 40% capric acid)	30 g/day (250 mL) for 6 months	Significant increase in total ketones post-intervention (mean + 0.416 mmol/L) in MCT group, 2 h after last administration of MCT	Cardiometabolic markers and peripheral inflammation	- Increase with time by group interaction for ketones, plasma capryc and caprylic acids, IL-8 - Main effect of time for insulin, TG, and NEFA - No effect on inflammatory markers (IL-6, IL-10, TNF-α) - No effect on BMI	- Unclear whether the observed changes were due to the chronic or acute ketosis - Unmonitored dietary modifications - Possible interference of participant’s medications on the response to the intervention
	Fortier et al. (2021) [45]	Double-blinded RCT (vs. placebo)	122	MCT (60% caprylic + 40% capric acid)	30 g/day (250 mL) for 6 months	Mean βHB increased to 0.4 (+/− 0.3) mmol/L 30 min after administration of MCT	Cognition Plasma ketone Metabolic profile	- Increase in plasma ketones - Improvement in episodic memory/executive function/language - Plasma ketones correlated positively with improvement in several cognitive tests - Post-interventional increase in plasma levels of glucose, cholesterol, aspartate transaminase, free caprylic, and capric acids - No change in BMI or body weight in either group	- No post-intervention wash-out period - No direct assessment of AD neuropathology biomarkers
	Fortier et al. (2019) [46]	Double-blinded RCT (vs. placebo)	52	MCT (60% caprylic + 40% capric acid)	30 g/day (250 mL) for 6 months	Mean βHB significantly increased to 0.543 (+/− 0.32) mmol/L	Metabolic rate of AcAc and glucose in the brain measured by PET with the 11C-AcAc and 18F-FDG tracers	- Increase in plasma AcAc and βHB - Increase in global brain CMR of AcAc and CMR of ketones (AcAc + βHB) in the active group - No change in whole or regional brain CMR of glucose in either group - Increase in net global brain energy uptake (CMR of Glucose + Ketones) - Improvement in processing speed and performance on the Visual Scan task of the Trail Making related to the increase in brain ketone uptake	Inadequately powered to assess the effect of Apoε4 status on cognitive outcomes
	Roy et al. (2021) [47]	Double-blinded RCT (vs. placebo)	33 (intervention: 17; placebo: 16)	MCT (60% caprylic + 40% capric acid)	30 g/day (250 mL) for 6 months	Tested +2 h post-prandial. Mean βHB increased significantly to 0.572 (+/− 0.32) mmol/L	Ketone metabolism by WM fascicles assessed by PET and diffusion imaging	- Increase in WM fascicle ketone uptake by 3-fold in active group - Improvement in processing speed associated with increase in ketone uptake in almost all WM - No variation of WM glucose uptake except for the posterior cingulate segment of the cingulum where a decrease is noted in both groups - No significant effect on WM structural properties	- Absence of myelin content assessment - Lack of specificity of radial diffusivity to myelin - Inability to segment WM hyperintensities and to establish whether both groups had similar cerebrovascular burden at baseline
	Rebello et al. (2015) [44]	Double-blinded RCT (vs. placebo)	6 (intervention: 3; placebo: 3)	MCT	56 g/day for 6 months	Tested +90 min post-prandial. βHB increased in Apoε4-, but was progressively less with measures repetition (0.15 mmol/L at week 24). βHB showed consistent increase in Apoε4+ status (0.54 mmol/L at week 24)	Blood βHB levels Cognition (ADAS-Cog/Trail Making Test/DSST)	- Slight increase in post-dose blood βHB levels, but was progressively less with measures repetition in the Apoε4- participant - Increase in ADAS-Cog in Apoε4-participant; overall decrease in ADAS-Cog in Apoε4 + participant - No improvements in Trail Making or DSST	- Only 4 participants completed the study - No statistical analysis performed
**AD**	Henderson et al. (2009) [33]	Double-blinded RCT (vs. placebo)	152 (intervention: 86; placebo: 66)	MCT (AC-1202, glycerin + caprylic acid)	20 g/day for 90 days	Tested +2 h post-prandial. Post-dose βHB levels significantly elevated to mean of 0.36–0.39 mmol/L for MCT group	Cognition (ADAS-Cog/ADCS-CGIC)	- Improvement in ADAS-Cog with more notable effects in Apoε4-subjects who were dosage compliant - No significant difference for the mean ADCS-CGIC or MMSE	- Low levels of blood βHB achieved - High drop-out rate, mainly due to gastrointestinal effects
	Henderson et al. (2020) [34]	Double-blinded RCT (vs. placebo)	413 (intervention: 208; placebo: 205)	MCT (AC-1204, 50% caprylic acid)	40 g/day for 6 months	Tested +1 h post prandial. Post-dose βHB levels significantly elevated to mean of 0.25 mmol/L for MCT group	ADAS-Cog11 score in Apoε4 non carriers	- No significant improvement in ADAS-Cog11, ADAS-CGIC, ADCS-ADL, CDIS, RUD-Lite and QOL-AD in Apoε4 non carriers - Significant degradation in MMSE in MCT group - Significant increase in βHB post dose - Higher incidence rate of discontinuations with MCT	- Low levels of blood βHB achieved - High drop-out rate, mainly due to gastrointestinal effects - Lack of disease progression probably influenced by enrolling primarily Apoε4-participants, with possible inclusion of suspected non-Alzheimer’s disease pathophysiology (SNAP) participants
	Croteau et al. (2018) [36]	Cross-over unrandomized controlled trial (C8C10 or C8)	15	MCT (60% caprylic + 40% capric acid or tricaprylin)	30 g/day (250 mL), each MCT formulation for 1 month	Tested +2 h post-prandial. Mean βHB significantly increased to 0.57 (+/− 0.27) mmol/L post-C8 -mean βHB significantly increased to 0.46 (+/− 0.19) mmol/L post-C8C10	Brain ketone and glucose uptake by PET	- Both MCTs significantly and similarly increased total brain energy metabolism by increasing ketone supply without affecting brain glucose utilization - Similar increase in plasma ketones for both MCTs	- Small sample size
	Ohnuma et al. (2016) [35]	Uncontrolled open-label trial	24	MCT (Axona, caprylic acid)	20 g/day for 3 months	Timing of testing post-intervention undisclosed. Mean βHB: 0.25 (+/− 0.20) mmol/L at M1 (then stable at M2 and M3)	- Tolerance of Axona in Japanese AD participants - Cognition (MMSE/ADAS-Jcog)	- Axona was well when the Axona Graduated Dosing Plan was implemented - No significant improvement in cognitive function (even in Apoε4- subjects)	Low levels of blood βHB achieved
	Xu et al. (2020) [37]	Cross-over double-blinded RCT (vs. placebo)	53	MCT (caprylic + capric acid)	17.3 g/day for 30 days	Timing of testing post-intervention undisclosed. Very weak mean βHB: 0.09 (+/− 0.07) mmol/L at M1	Cognition (ADAS-Cog-C)	- Improvement in ADAS-Cog-C in Apoε4- participants only - Increase in blood concentrations of TG, HDL-c, βHB and AcAc - Improvement in the ADL scale	- Body height/body weight/BMI were not included - Achieved blood βHB levels were very weak; Metabolic changes in peripheral blood do not directly correspond to the metabolic changes in brain tissue
	Chan et al. (2017) [38]	Double-blinded RCT (vs. placebo)	40 (intervention: 20; placebo: 20)	MCT (cold pressed coconut oil)	30 mL/day during 2 weeks then 60 mL/day during 22 weeks	Not measured	- Cognition (MMSE/CDIS) and behavior (NPI-Q) - Adverse effects on blood parameters and electrocardiogram	- No significant difference in all parameters between both groups - No difference in MMSE and NPI-Q from baseline to 6 months within each group - Significant improvement in CDIS with placebo - Alanine aminotransferase and TG increased with MCT - ASAT increased with placebo, urea and uric acid decreased with placebo - No significant change in electrocardiogram	- Small sample size - High rate of drop outs - No blood levels of βHB measured - No stratification with Apoε4 status
	De la Rubia Orti et al. (2018) [39]	Double-blinded RCT (vs. placebo)	44 (intervention: 22; placebo: 22)	MCT (coconut oil)	40 mL/day for 21 days	Not measured	Temporal orientation, visuospatial abilities, semantic and episodic memory	- Improvements in temporal orientation and episodic and semantic memory - Positive effect is more evident in women with mild-moderate state	- Small sample size - Short duration of exposition to intervention - No blood levels of βHB measured - No stratification with Apoε4 status
	Juby et al. (2022) [40]	Cross-over double-blinded RCT (vs. placebo)	20	MCT	42 g/day or maximum tolerated for 4 months, then open label for 7 months	Tested in fasting condition. Baseline βHB: 0.19 mmol/L, then at study completion 0.22 mmol/L	Cognition (Cognigram tests/MMSE/MOCA)	- Significant improvement in Cognigram1 in participants who received 11 months of uninterrupted MCT - No significant difference in Cognigram2 - No significant impact of weight, height, hip circumference or Apoε4 status on cognitive assessments responses to MCT - Significant impact of BMI and waist circumference on cognitive assessments responses to MCT	- Small sample size - Wide spectrum of ages and AD disease states - Lack of continuous βHB monitoring - Difficulty reaching maximum dose due to side effects or missing lunchtime dose - Unpowered to evaluate impact on MMSE and Cognigram
	Torosyan et al. (2018) [41]	Double-blinded RCT (vs. placebo)	16 (intervention: 14; placebo: 2)	MCT (caprylidene)	40 g/day for 45 days	Not measured	Evaluation of acute and long-term effects of caprylidene on regional CBF	- No significant acute or long-term changes in regional CBF across all subjects - Significant long-term increase in regional CBF in the left superior lateral temporal cortex, anterior cerebellum, left inferior temporal cortex and hypothalamus in Apoε4-subgroup - No significant long-term increase in regional CBF observed (in any region)	- Small sample size particularly the placebo group (2 participants) - Results not compared with placebo group - No control for the regular diet of the participants - No blood levels of βHB measured
	Ota et al. (2019) [42]	Study 1: Cross-over double-blinded RCT (vs. placebo) Study 2: Open-label study	Study 1 and 2: 20	MCT (Ketonformula^®^, 50 g of this formula contains 20 g MCT)	Study 1: single dose of 50 g Ketonformula Study 2: 50 g/day Ketonformula during 12 weeks	Study 1: Tested before consumption and then 2 h later with significant increase to 0.47 (+/− 0.29) mmol/L Study 2: Tested before consumption in weeks 4–8–12 with no significant increase	Effect of single (Study 1) and chronic (Study 2) administration of MCT on cognitive function	Study 1: - No significant difference between scores in any cognitive test Study 2: - Significant improvements in the DSST and immediate logical memory test when compared the cognitive scores at 12 weeks with those at baseline - Significant improvements in the immediate and delayed logical memory tests between the scores at 8 weeks and baseline	- Small sample size - Open-label trial cannot rule out placebo effects - Effect of Apoε4 not assessed - Low levels of blood βHB achieved
	Fernando et al. (2023) [43]	Double-blinded RCT (vs. placebo)	120 (intervention: 60; placebo: 60)	MCT (virgin coconut-oil)	30 mL/day for 24 weeks	Not measured	- Cognition (MMSE/CDIS) - Impact of Apoε4 genotype on the changes in cognition	- No significant difference in cognitive scores between groups post-intervention - Significant improvement in MMSE among Apoε4 carriers who received virgin coconut oil compared to non-carriers	- High drop-out rate (30%), similar between groups - MMSE and CDIS are non-specific cognitive outcomes and may not reflect improvements in specific cognitive domains - Behavioral and psychological symptoms of dementia were not assessed - Energy and fat intake in the background diet was higher than the habitual diet and may have influenced the results - No blood levels of βHB measured
**MCI and AD**	Reger et al. (2004) [32]	Cross-over double-blinded RCT (vs. placebo)	20	MCT	40 mL single dose	Tested at +90 min and +120 min post-prandial. Mean βHB increased to 0.54 (+/− 0.32) mmol/L at + 90 min and remained stable in Apoε4- participants. Mean βHB increased to 0.43 (+/− 0.16) mmol/L at + 90 min and again to 0.68 (+/− 0.36) mmol/L at +120 min in Apoε4+ participants	Neuropsychological testing (ADAS-Cog/Stroop Color Word Test/paragraph recall)	- Improvement in ADAS-Cog in Apoε4-participants only - Higher ketone values associated with greater improvement in paragraph recall in all subjects (Apoε4 − and +) - Improvement for the Stroop Color Word Test - Higher blood βHB levels at the end of the study in Apoε4+ subjects	- Small sample size - Assessment of the impact of acute ketosis only - Low level of blood βHB reached
**PD**	Norwitz et al. (2020) [49]	Cross-over single-blinded RCT (vs. placebo)	15	Ketone ester (DeltaG)	25 mL single dose	Mean βHB increased significantly to 3.5 (+/− 0.3) mmol/L within 30 min of ketone ester consumption	Length of time participants could sustain an 80 revolutions per minute (rpm) cycling cadence	- Improvement in endurance performance - Increase in blood βHB levels - Decrease in RER in ketone group - No variation in other cardiorespiratory parameters (O_2_ consumption/CO_2_ consumption/energy expenditure) or lactate	- Intrapersonal variability of Parkinson’s disease symptomatology - “Off” medication state - Lack of power of the study to reach statistical significance for the secondary endpoints
**Multiple Sclerosis**	Platero et al. (2020) [50]	Double-blinded RCT (vs. placebo)	51 (intervention: 27; placebo: 24)	MCT (extra-virgin coconut oil)	60 mL/day for 4 months	Tested after an overnight fast. Median βHB: 0.05 (range 0.33) mmol/L	- State and trait anxiety (STAI) - Functional disability (EDSS) - Serum IL-6	- Decrease in STAI - Decrease in EDSS - Decrease in IL-6 in both groups - Decrease in BMI in both groups	Impossible to distinguish the individual impact of each of the treatment components (EGCG, coconut oil, mediterranean diet) on the different outcomes evaluated
	Platero et al. (2021) [51]	Double-blinded RCT (vs. placebo)	51 (intervention: 27; placebo: 24)	MCT (extra-virgin coconut oil)	60 mL/day for 4 months	Tested after an overnight fast. Median βHB: 0.05 (range 0.33) mmol/L	Cortisol activity related to fat loss and depression	- No variation in cortisol - Decrease in levels of depression and abdominal fat - Increase in albuminemia - Decrease in total weight in both groups - Muscle percentage increased with MCT and decreased with placebo	No evaluation of variations in ACTH
**Episodic migraine**	Putananickal et al. (2022) [52]	Cross-over double-blinded RCT (vs. placebo)	41	Ketone salt (Ergomax) (Ca-βHB (9 g)/Mg-βHB (9 g))	18 g/day for 3 months	Tested +40 min post-prandial. Median βHB: 0.40 (interquartile-range [0.30, 0.50])	Number of migraine days in the last four weeks of treatment (adjusted for baseline)	- No decrease in migraine days - No decrease in average migraine intensity - No variation in consumption of acute migraine medication - No impact on MIDAS and HIT-6	- Ketone salt was a chiral molecule with two enantiomers, and L-3-hydroxybutyrate is not significantly metabolized into energy intermediates - Insufficient blood βHB reached - Natural fluctuations in migraine - Heterogeneity of migraineurs
**Drug-** **resistant epilepsy**	Rasmussen et al. (2023) [53]	Uncontrolled open-label trial	9	MCT (50% caprylic + 30% capric acid)	112 g/day (target dose) for 3 months	Not measured. Patients were screened for urinary ketones (+ in 5/6 individuals at some point during the trial)	Reduction in number of seizures	- 5/6 participants who completed the trial recorded a significant reduction in number of seizures - Urinary ketones were + in 5/6 participants at some point during the trial - Reported adverse events: mild nausea, stomachache and loose stools	- Very small number of participants who completed the study (6) - Uncontrolled trial - Limited follow-up time - Wide variability in the doses of MCT administered to subjects depending on their tolerance

Abbreviations: 18F-FDG: [18F]-fluorodeoxyglucose; AcAc: acetoacetate; ACTH: adrenocorticotropic hormone; AD: Alzheimer’s disease; ADAS-Cog: Alzheimer’s Disease Assessment Scale Alzheimer’s-cognitive; ADAS-Cog-C: Alzheimer’s Disease Assessment Scale–Cognitive Subscale; Chinese version; ADAS-Jcog: Alzheimer’s Disease Assessment Scale (Japanese version) cognitive subscale; ADCS-ADL: Alzheimer’s Disease Cooperative Study–activities of daily living; ADCS-CGIC: Alzheimer’s Disease Cooperative Study–Clinical Global Impression of Change; ADL: activities of daily living; Apoε4: apolipoprotein ε4; BMI: body mass index; CDIS: clock drawing interpretation scale; CMR: cerebral metabolic rate; EDSS: Expanded Disability Status Scale; EGCG: epigallocatechin gallate; HDL-c: high-density lipoprotein cholesterol; HIT-6: Headache Impact Test-6; IL: interleukin; MCI: mild cognitive impairment; MCT: medium-chain triglyceride; MIDAS: Migraine Disability Test; MOCA: Montreal Cognitive Assessment; NPI-Q: Neuropsychiatric Inventory Questionnaire; NEFA: non-esterified fatty acids; PD: Parkinson’s disease; PET: positron emission tomography; QOL-AD: quality of life–Alzheimer’s disease; RCT: randomized controlled trial; RER: Respiratory Exchange Ratio; RUD-Lite: resource utilization in dementia-lite; SNAP: suspected non-Alzheimer’s disease pathophysiology; STAI: State-Trait Anxiety Inventory; TG: triglyceride; TNF-α: tumor necrosis factor α; WM: white matter; βHB: β-hydroxybutyrate.

**Table 4 nutrients-17-03125-t004:** Overview of studies on psychiatric disorders.

**PTSD**	Youssef et al. (2022) [54]	Double-blinded RCT (vs. placebo)	21 (intervention: 11; Placebo: 10)	Ketone salt	14 g/day for 6 weeks	Not measured	PTSD symptoms (PCL-5)	- No difference in PCL-5 medians between both groups - Decrease in median PCL-5 scores from pre-test to post-test in active group	- Small sample size - No blood levels of βHB measured - Important intragroup variability of PTSD severity as randomization was not based on PTSD severity

Abbreviations: RCT: randomized controlled trial; PCL-5: PTSD Checklist for DSM-5; PTSD: post-traumatic stress disorder; βHB: β-hydroxybutyrate.

**Table 5 nutrients-17-03125-t005:** Overview of studies on metabolic disorders.

Condition	Reference (Year)	Study Design	Participants (n)	Type of Exogenous Ketone	Dose of Exogenous Ketone	Blood βHB Level	Primary Outcomes	Main Findings	Main Limitations
**Prediabetes**	Bharmal et al. (2021) [55]	Cross-over double-blinded RCT (vs. placebo)	18	Ketone ester (DeltaG)	395 mg/kg single dose	Tested +30 min post-prandial. Mean βHB increased significantly to 3.47 (+/− 0.22) mmol/L	Blood glucose	- Decrease in AUC (0–150 min) for glucose - Increase in AUC (0–150 min) for insulin and C-peptide and GIP - No variation for amylin and glucagon and GLP-1	- No evaluation of the effect of ketone ester supplementation on other glucoregulatory mechanisms - Absence of other nutritional stimulants or concomitant food components
	Bharmal et al. (2021) [56]	Cross-over double-blinded RCT (vs. placebo)	18	Ketone ester (DeltaG)	395 mg/kg single dose	Tested +30 min post-prandial. Mean βHB increased significantly to 3.47 (+/− 0.22) mmol/L	Modulation of intra-abdominal fat distribution on the effect of exogenous ketones on glucoregulatory peptides	- βHB increased in both high/low adiposity groups - Increase in total AUCs for insulin and C-peptide in participants with high intra-pancreatic fat deposition, skeletal muscle fat deposition, subcutaneous fat volume and also in participants with low visceral fat volume and intra-hepatic fat deposition - Increase in total AUC for GIP in participants with high intra-pancreatic fat deposition, skeletal muscle fat deposition, visceral fat volume and subcutaneous fat volume - Total AUC for GLP-1 was not associated with any adiposity phenotype	- Sample size was calculated based on the change in plasma glucose following exogenous ketosis - No state-of-the-art indices of lipolysis - Analysis of a limited set of gut hormones involved in lipid metabolism and the maintenance of glucose homeostasis
	Kimita et al. (2021) [57]	Cross-over double-blinded RCT (vs. placebo)	18	Ketone ester (DeltaG)	395 mg/kg single dose	Tested +30 min post-prandial. Mean βHB increased significantly to 3.47 (+/− 0.22) mmol/L	Plasma levels of markers of iron metabolism	No significant effect on circulating levels of markers of iron metabolism	- No evaluation of long-term effects of exogenous ketosis on iron metabolism - Many other markers of iron metabolism not evaluated
	Liu et al. (2022) [58]	Cross-over double-blinded RCT (vs. placebo)	18	Ketone ester (DeltaG)	395 mg/kg single dose	Tested +30 min post-prandial. Mean βHB increased significantly to 3.47 (+/− 0.22) mmol/L	Plasma lipid profile	- Decrease in remnant cholesterol and TG - No variation in total cholesterol, LDL-c or HDL-c - Decrease in remnant cholesterol and TG in participants with high saturated fat intake - No variation in all outcomes measured in participants with low saturated fat intake	- Inadequately powered - No investigation of the size and number of LDL particles or apolipoproteins - Remnant cholesterol was calculated from the standard lipid panel rather than measured - No evaluation of the effect of acute ketosis on FFA - Interference of genetic dyslipidemia not excluded
	Liu et al. (2023) [59]	Cross-over double-blinded RCT (vs. placebo)	18	Ketone ester (DeltaG)	395 mg/kg single dose	Tested +30 min post-prandial. Mean βHB increased significantly to 3.47 (+/− 0.22) mmol/L	Objective and subjective parameters of appetite regulation (acylated ghrelin, peptide YY, and hunger)	- No variation was noted in the AUCs for ghrelin, peptide YY, and hunger - No impact of eating behaviors on the effect of exogenous ketosis on both objective and subjective parameters of appetite regulation	- Non-isocaloric placebo - Evaluation of only one aspect of subjective appetite assessment and only classic appetite-related hormones
	Charles et al. (2023) [60]	Cross-over double-blinded RCT (vs. placebo)	18	Ketone ester (DeltaG)	395 mg/kg single dose	Tested +30 min post-prandial. Mean βHB increased significantly to 3.47 (+/− 0.22) mmol/L	Plasma levels of asprosin and leptin	No variation in plasma levels of asprosin and leptin, even after stratification by abdominal fat phenotypes	Long-term effects of exogenous ketosis on plasma levels of asprosin and leptine not evaluated
	Charles et al. (2024) [61]	Cross-over double-blinded RCT (vs. placebo)	18	Ketone ester (DeltaG)	395 mg/kg single dose	Tested +30 min post-prandial. Mean βHB increased significantly to 3.47 (+/− 0.22) mmol/L	GDF-15 levels	- No significant difference in the AUC for GDF-15 - No significant impact of eating behaviors in the effect of ketone ester on GDF-15 levels	- Trial was not powered to investigate changes in GDF-15 levels - No evaluation of the effect of long-term ketone ester consumption on circulating levels of GDF-15
	Nakagata et al. (2021) [62]	Cross-over double-blinded RCT (vs. placebo)	9	Ketone ester	482 mg/kg single dose	Tested +90 min post-prandial. Significant increase in mean βHB: 2.4 (+/− 0.7) mmol/L	Blood glucose levels during the 75 g OGTT	- Decrease in AUC of glucose - Increase in AUC of insulin during the first half of OGTT with no variation in AUC of C-peptide - No variation of glucagon, FFA, Matsuda index or insulinogenic index	- Small sample size - All the participants were East-Asians individuals and they have a limited innate capacity for insulin secretion
**Type II Diabetes Mellitus**	Jensen et al. (2020) [67]	Cross-over double-blinded RCT (vs. placebo)	18	Ketone salt (Na-DL-3-hydroxybutyrate), intravenous	0.22 g/kg/h during approximately 165 min	Significant increase in mean βHB levels throughout neurocognitive assessment (time: +120 to 165 min): 2.4 (+/− 0.6) mmol/L	Cognitive performance (global cognitive composite score, estimated by the average of 4 domains: verbal memory, working memory and executive functions, psychomotor speed, and sustained attention	No significant difference in global cognitive composite score, even if performance of WAIS-LNS (working memory) improved significantly with ketone infusion	- Small sample size - Study doesn’t explore if any changes in cognitive outcomes were related to cerebral metabolism - WAIS-LNS only does not provide a full evaluation of working memory - Study limited to participants without dementia or mild cognitive impairment, with good glycemic control as well as absence of any severe diabetic complications - Racemic mixture of D- and L-βHB but only the D-isoform was measured
	Monteyne et al. (2024) [66]	Cross-over double-blinded RCT (vs. placebo)	10	Ketone ester	500 mg/kg single dose	Tested +60 min post-prandial. Significant increase with peak βHB: 4.3 (+/− 1.2) mmol/L	Reduction in post-prandial blood glucose	- Significant reduction of 2 h and 4 h post-prandial glucose concentrations, mainly as a result of a decrease in the 2 h rate of glucose appearance following meal ingestion - No effect on post-prandial endogenous glucose production - No significant difference in plasma insulin - Suppressed plasma NEFA concentrations after ketone ester ingestion	- Small sample size - Post-prandial period evaluated was in the context of fast breaking and it is unclear whether the plasma blood glucose-lowering effect would be retained for subsequent meals and how it might be modulated by circadian fluctuations in metabolism - Trial unpowered to determine if biological sex has the potential to moderate the observed effect
	Oliveira et al. (2024) [65]	Cross-over double-blinded RCT (vs. placebo)	18	Ketone ester	0.3 g/kg single dose	Tested +60 min post-prandial. Significant increase with peak βHB: 1.8 (+/− 0.6) mmol/L	Determine whether acute ingestion of ketone ester supplement influenced hunger, fullness and food intake	- No significant difference over time in self-reported hunger, fullness or food intake - Significantly higher energy intake from protein in the ketone condition	- Small sample size - Possible “lingering” effects of the glucose-lowering medication that participants were on before participating in study - No evaluation of long-term effects of ketone ester ingestion on appetite related variables and resultant food intake
	Falkenhain et al. (2024) [64]	2 cross-over double-blinded RCTs (vs. placebo)	Study 1: 18 Study 2: 15	Ketone ester (DeltaG for study 1, KetoneAid for study 2)	0.3 g/kg single dose for Study 1, 45 g/day (90 mL) for 14 days for Study 2	Study 1: Tested at +60 min post-prandial, significant increase in mean βHB: 1.8 (+/− 0.6) mmol/L; Study 2: Tested at +30 min post-prandial, significant increase in mean βHB: 1.8 (+/− 0.7) mmol/L	Study 1: Plasma glucose Study 2: Serum fructosamine	Study 1: - No difference in plasma glucose, hemodynamic measures, gastro-intestinal distress and PBMC metabolites - Decreased in NEFA; Increase in insulin with no difference for C-peptide Study 2: - No variation in serum fructosamine, anthropometric and fasting metabolic outcomes - Increase in reported gastrointestinal symptoms	Study 1: Discontinuation of glucose-lowering medication only 1 day before each study visit Study 2: Continuation of glucose-lowering medications possibly inducing interference with study outcomes
	Soto-Mota et al. (2021) [63]	Uncontrolled open-label trial	23	Ketone ester	75 g/day (75 mL) for 4 weeks	Tested +30 min post-prandial. Significant increase in βHB with range from 3.1 (+/− 0.5) to 3.8 (+/− 0.7) mmol/L	Safety, tolerability and effects on glycemic control	- Decrease in fructosamine, HbA1c and mean daily glucose - No variation in total body weight, total body fat, visceral fat, TG, total cholesterol, HDL-c, Apolipoprotein B, CRP, HOMA–IR and 10-year fatal coronary heart disease risk - Mild and infrequent adverse events	- Small sample size - Uncontrolled trial - Limited follow-up time - No measure of participant’s food consumption
**Obesity**	Myette-Côté et al. (2019) [68]	Cross-over double-blinded RCT (vs. placebo)	15	Ketone ester (DeltaG)	482 mg/kg single dose	Peak βHB: 3.4 mmol/L. Tested +90 min post-prandial	Capacity to lower blood glucose concentration	- Decrease in glycemic and NEFA response to OGTT - Increase in oral glucose insulin sensitivity index - No changes in flow or shear patterns in CCA - Decrease in MAP and increase in HR - No variation of TG and lactate levels or for their AUCs	- Non-isocaloric placebo - Assessment only of the effects of acute exogenous ketosis - Trial have not been powered to assess differences in gender - Participants were not taking any antidiabetic or lipid lowering medications
	Neudorf et al. (2020) [69]	Cross-over double-blinded RCT (vs. placebo)	15	Ketone ester (DeltaG)	482 mg/kg single dose	Tested +60 min post-prandial. Mean βHB increased significantly to 2.96 (+/− 0.91) mmol/L	Effect on NLRP3 activation	- No impact on caspase-1 activation in unstimulated (no LPS, basal caspase-1 activation) or stimulated (with LPS) monocytes - No variation in secretion of IL-1β or TNF-α or IL-6 in stimulated monocytes	- Test of a single NLRP3 activator - Limited exposure time to exogenous ketosis
	Kanta et al. (2025) [72]	Study 1: Cross-over single-blinded RCT (vs. LCT) Study 2: Cross-over double-blinded RCT (vs. LCT) but included only healthy participants (excluded from analysis)	16 (with 8 healthy controls)	MCT (caprylic acid)	35 g single dose during test days (pretest and posttest), then increasing doses from 10 g 2/day to 30 g 2/day during 7 days	Significant increase with Peak βHB: 0.7 (+/− 0.1) mmol/L. Tested +90 min post-prandial.	Post-prandial energy metabolism	- Significant increase in metabolic rate with obese participants on MCT - Significant reduction of blood glucose over 5 h with obese participants on MCT - Significant and transient increase in plasma insulin and glucagon levels with obese participants on MCT - These effects on metabolic rate and glycemia were preserved in individuals with obesity and sustained after 8 days of daily supplementation	- Small sample size - Habitual dietary patterns and physical activity levels during the trial were only based on self-reporting - Only male adults involved - Short duration of exposition
	Yu et al. (2025) [73]	Cross-over double-blinded RCT (vs. placebo)	40 (with 20 healthy controls)	Ketone ester (HVMN)	395 mg/kg single dose	Significant increase with βHB level raised by 0.92 mmol/L compared to placebo group	Metabolic and neurocognitive indicators (i.e., PFC connectomes (causal density) and cognitive interference)	*Linear mixed models analysis:* - Significant effects were observed for trial and assessment time in βHB, glucose, causal density, and cognitive interference but not for group factors (obese vs. healthy) - Effects of ketone ester on prefrontal connectomes were more enduring in healthy participants - Elevated βHB level improved cognitive function through decreasing glucose level and increasing causal density - Significant increase in levels of blood BHB and prefrontal connectomes and decrease in levels of glucose and cognitive interference, regardless of weight status with ketone ester	- Small sample size - Non representative of elderly population or participants with higher BMIs - Includes mainly overweight and not concretely obese participants - Evaluate only acute effects of ketone ester supplementation (no evaluation of long-term metabolic and cognitive changes) - Benefits observed post-supplementation might not sustain or could evolve beyond the 90 min observation window
	Walsh et al. (2021) [71]	Cross-over double-blinded RCT (vs. placebo)	15	Ketone ester (DeltaG)	36 g/day for 2 weeks	Tested +15 min post-prandial. Significant increase in mean βHB to 1.8 (+/− 1.3) mmol/L	Post-prandial glycemia, vascular function and inflammatory markers	- Decrease in post-prandial glucose AUC and in 24-h mean glucose - Improvement in endothelial function (%FMD) - Reduction of NLRP3 activation - No variation in circulating cytokines - No differences in PROMs: hunger/fullness/satisfaction/capacity to eat more	- Limited follow-up time - Participants had no impaired glucose tolerance and therefore these results may not be applicable to subjects with dysglycemia - Continuous glucose monitors measure interstitial glucose which can be impacted by changes in subcutaneous blood flow - No evaluation of tissue-specific effects of βHB on local inflammatory pathways
	Walsh et al. (2021) [70]	Cross-over double-blinded RCT (vs. placebo)	15	Ketone ester (DeltaG)	36 g/day for 2 weeks	Tested +15 min post-prandial. Significant increase in mean βHB to 1.8 (+/−1.3) mmol/L	CBF, BDNF, and cognitive function assessed by DSST/Stroop task/TST	- Increase in CBF and improvement in DSST - Positive correlation of improvements in DSST and cerebrovascular outcomes - No variation in BDNF, Stroop task, TST, fasting plasma glucose, insulin, C-peptide, NEFA, body weight or body composition - Similar treatment adherence	- Limited follow-up time - Most participants were women (2/3) - CBF measured only at rest - Trials have not been powered to detect changes in BDNF
**VLCADD**	Bleeker et al. (2020) [74]	Cross-over double-blinded RCT (vs. placebo)	5	Ketone ester	395 mg/kg single dose	Tested +30 min post-prandial. Significant increase in βHB to 2.0 (+/− 0.2) mmol/L	Quadriceps phosphocreatine, inorganic phosphate and pH dynamics during exercise and recovery assayed by in vivo ^31^P-MR	- Decrease in inorganic phosphate/phosphocreatine ratio in exercising leg muscle - Increase in blood βHB then significant drop during the second bout of cycling in active arm - No variation in blood glucose during the entire protocol in both arms then significant drop during the second bout of cycling in control arm - Intervention well tolerated by participants - Transient increase in plasma insulin levels in both arms - No variation in plasma lactate levels - Decrease in total concentration of long-chain acylcarnitines - Decrease in FFA - Increase in plasma acetylcarnitine - Effective βHB muscle intake - Blunt the increase in intramuscular glycolytic intermediates during exercise seen after carbohydrate ingestion	- Small sample size - Unavailability of data for some outcomes
**McArdle Disease**	Løkken et al. (2022) [75]	Cross-over double-blinded RCT (vs. placebo)	12	Ketone ester (HVMN)	395 mg/kg single dose	Tested +25 min post-prandial (immediately before exercise). Significant increase in mean βHB to 3.3 (+/− 1.3) mmol/L	Exercise capacity as indicated by heart rate response to exercise	- No improvement in exercise capacity - No variation in perceived exertion - Increase in relative ketone bodies-oxidation rates in active group - Decrease in plasma glucose, FFA, lipolytic rate and glucose rate of appearance - Increase in plasma insulin levels - No variation in plasma lactate, pyruvate, and ammonia in diseased subgroup unlike the healthy group (decrease) - Decrease in concentrations of amino-acids in diseased subgroup	- Small sample size - Only evaluation of acute effects of exogenous ketosis
	Valenzuela et al. (2024) [76]	Cross-over double-blinded RCT (vs. placebo, vs. CHO drink)	15 (with 7 healthy controls)	Ketone ester (KetoneAid)	30 g single dose	Test +30 min post-prandial (immediately before exercise). Significant increase in mean to 3.7 (+/− 0.9) mmol/L	Exercise capacity (or performance) assessed by a constant-load then maximal ramp test	- Significant and similar increase in βHB in McArdle/healthy controls * Constant load test: * - Significantly lower power output in submaximal constant tests and gross-efficiency in McArdle participants with ketone ester compared to healthy controls - Significantly higher HR, rating of perceived pain or exertion and RER in McArdle participants with ketone ester compared to healthy controls - Significantly lower HR with CHO compared with ketone esters in McArdle participants * Maximal ramp test: * - Significantly lower VT, PPO, peak VO2, peak RER in McArdle participants with ketone ester compared to healthy controls - Significantly higher VT, PPO, and peak VO2 with CHO compared with ketone esters in McArdle participants	- Small sample size - No objective determination of muscle glycogen levels in healthy controls group - Other potential interacting variables uncontrolled as menstrual cycle phase or contraceptive use - Only evaluation of acute effects of exogenous ketosis

Abbreviations: AUC: area under the curve; BDNF: brain-derived neurotrophic factor; CBF: cerebral blood flow; CCA: common carotid artery; CHO: carbohydrate; CRP: C-reactive protein; DSST: Digit Symbol Substitution Task; FFA: free fatty acid; FMD: flow-mediated dilation; GDF-15: Growth Differentiation Factor-15; GIP: glucose-dependent insulinotropic polypeptide; GLP-1: glucagon-like peptide-1; HDL-c: high-density lipoprotein cholesterol; HOMA-IR: Homeostatic Model Assessment for Insulin Resistance; HR: heart rate; IL: interleukin; LCT: long-chain triglycerides; LDL-c: low-density lipoprotein cholesterol; LPS: lipopolysaccharide; MAP: mean arterial pressure; MCT: medium-chain triglycerides; NEFA: non-esterified fatty acid; NLRP3: NOD-like receptor pyrin-domain containing 3; OGTT: Oral Glucose Tolerance Test; PBMC: peripheral blood mononuclear cell; PFC: prefrontal cortex; PPO: peak power output; PROMs: Patient-Reported Outcome Measures; RCT: randomized controlled trial; TG: triglyceride; RER: Respiratory Exchange Ratio; TNF-α: tumor necrosis factor α; TST: Task Switching Task; VLCADD: very long-chain acyl-CoA dehydrogenase deficiency; VT: ventilatory threshold; WAIS-LNS: Wechsler Adult Intelligence Scale–Letter–Number Sequencing; βHB: β-hydroxybutyrate.

**Table 6 nutrients-17-03125-t006:** Overview of studies on cardiovascular disorders.

Condition	Reference (Year)	Study Design	Participants (n)	Type of Exogenous Ketone	Dose of Exogenous Ketone	Blood βHB Level	Primary Outcomes	Main Findings	Main Limitations
**Chronic HFrEF**	Monzo et al. (2021) [77]	Uncontrolled open-label trial + observational study (excluded from analysis). 11 HFrEF participants and 6 non-HFrEF subjects	17	Ketone ester (HVMN)	25 g single dose	Tested +80 min post-prandial. Mean βHB significantly increased by 12.9-fold (concentration not reported)	Myocardial substrate utilization	- No adverse events - Reduction in venous pH and kalemia - Decrease in FFA and FFA FE% in HFrEF participants - Higher lactate FE% in HFrEF participants - Greater increase in βHB FE% in HFrEF participants - βHB FE% directly correlated LV mass, LV end-diastolic dimension, transmyocardial release of BNP and inversely with LVEF - No correlation between βHB FE% and FFA or glucose FE% - Positive correlation between βHB and lactate FE% - Lower βHB FE% in diabetic subjects	- Small sample size - Absence of myocardial blood measure hinders the absolute quantification of substrate utilization - Mild myocardial dysfunction that might influence metabolism is not excluded in non-HFrEF participants (indication for a cardiac pacing) - Administration of fentanyl during intraprocedural sedation may affect cardiac metabolism at high doses but participants received only slight sedation with the same doses
	Nielsen et al. (2019) [78]	3 cross-over single-blinded RCTs (vs. placebo) and 1 dose response study. Study 4 is performed in age-matched volunteers. Study 2 is a dose-response study	Study 1: 16 Study 2: 8 Study 3: 10 Study 4: 10	Ketone salt (Na-βHB, racemic mixture 50/50 D/L), intravenous	Study 1: 0.18 g/kg/h for 3 h Study 2: 0.045 g/kg/h for 2 h then 0.09 g/kg/h for 2 h Study 3 and 4: 0.18 g/kg/h for 3 h	Study 1: Tested H + 3 in the interventional group, significant increase in mean βHB to 3.3 (+/− 0.4) mmol/L Study 2: Tested H + 2 for each intervention, mean βHB: 0.7 (+/− 0.1) mmol/L at an infusion rate of 0.045 g/kg/h and mean βHB: 1.6 (+/− 0.3) mmol/L at an infusion rate of 0.09 g/kg/h Study 3 and 4: Tested H + 3, mean βHB 3.4 (+/− 0.6) mmol/L in the interventional group with no difference between HFrEF (Study 3) and healthy individuals (Study 4)	Study 1 and 2: CO measured by thermodilution Study 3 and 4: MEE, MVO2 and MBF assessed by 11C-acetate PET	Study 1: - Improvement in CO and SVO2 by an increase in SV and HR - Decrease in CVP, PCWP, SVR, PVR and kalemia - Increase in LVEF, GLS, lactate and pH - No variation in MAP or mPAP Study 2: - Variations in outcomes similar to those observed in Study 1, but were gradual - No variation for kalemia and lactatemia Study 3 and 4: - Increase in LV external work and MVO2 - No variation in MEE - Increase in MBF with an increment slightly higher in age-matched volunteers Study 1 to 4: - No variation in BNP, troponin T, noradrenalin and insulin levels - Decrease in FFA - Decrease in adrenaline levels in age-matched volunteers	- Potential carry-over effect - Investigator who performed CO assessment by thermodilution (for Study 1 and 2) was not blinded to the randomization - Coadministration of insulin during the interventions
**Chronic HFpEF**	Gopalasingam et al. (2024) [79]	Cross-over double-blinded RCT (vs. placebo)	28 (at randomization)	Ketone ester (KetoneAid)	25 g 4/day for 2 weeks	Significant increase in mean βHB of the 4-h observation period at rest to approximately 1 mmol/L	CO measured by thermodilution at rest and cardiopulmonary exercise	* During the 4 h rest period following ketone ester intake: * - Significant increase in CO, HR, average and septal e’ velocity and maximum velocity, - Significant decrease in PCWP and SVR - Significant rightward shift of the EDPVR in the LV with decrease in the calculated LV chamber stiffness beta * At peak cardiopulmonary exercise following ketone ester intake: * - Significant increase in CO, SV, and arterial oxygen saturation, - Significant decrease in PCWP - Significant decrease in the pressure-flow relationship	- Carry-over effects risk - Evaluation of only 2 weeks of exposition to ketone ester - Participants were mainly male with ischemic heart disease and without AF - Participants had only early to mild stage HFpEF - Some participants had LVEF between 40–49% which is not usually classified as HFpEF - CO measure with thermodilution may have limitation in accuracy in patients with AF
**Cardiogenic Shock**	Berg-Hansen et al. (2023) [80]	Cross-over double-blinded RCT (vs. placebo)	13	Ketone ester	500 mg/kg single dose	Significant rise in mean βHB to 2.9 (+/− 0.5) mmol/L during the 3-h treatment period	CO measured by thermodilution and expressed as AUC	- Improvement in CO driven by higher SV - No HR acceleration - Increase in CPO, SVO2, PrSO2, LVEF, GLS, S’max, LVOT VTI and TAPSE - Decrease in SVR, CVP, PCWP, LV filling pressure, pH, partial pressure of CO_2_, bicarbonatemia, kalemia, and blood glucose - No variation in MAP, mPAP, PVR, CrSO2, diuresis, accumulated doses of furosemide, insulin, lactate, FFA, and BNP levels - No difference in safety criteria with similar troponin T, VIS, and MAP	- Small sample size - Potential carry-over effect - High risk of gastroparesis during cardiogenic shock potentially causing a delayed biological response to ketone ester administration
**PH**	Nielsen et al. (2023) [81]	Cross-over double-blinded RCT (vs. placebo)	20	Ketone salt (Na-βHB, racemic mixture 50/50 D/L), intravenous	0.18 g/kg/h for 2 h	Tested +30 min post-intervention. Significant rise in mean βHB to 3.4 mmol/L	CO measured by thermodilution	- Improvement in CO due to higher SV and HR, with increase in SVO2 - Increase in mPAP due to an increase in systolic PAP and stable diastolic PAP - Improvement in contractile function measured by right ventricular systolic tricuspid annular velocity and in measurements of LV function (LVEF, SV, and GLS) - Increase in lactate, pH, sodium, and insulin - Decrease in PVR, RAP in PH subgroup, and magnesium - No variation in PCWP, pulmonary vascular capacitance, strain (RV-free wall, RV-global longitudinal strain), TAPSE and RV-fractional area	- Only evaluation of acute effects of exogenous ketosis - Potential carry-over effect - Dissociation of Na-3-OHB into 3-OHB- and Na+ enables 3-OHB- to act as a weak base and to increase pH possibly leading to hemodynamic effects

Abbreviations: 3-OHB: 3-hydroxybutyrate; AF: atrial fibrillation; AUC: area under the curve; BNP: brain natriuretic peptide; CO: cardiac output; CPO: cardiac power output; CrSO2: cerebral regional tissue saturation; CVP: central venous pressure; EDPVR: end-diastolic pressure-volume relationship; FE%: fractional extraction; FFA: free fatty acid; GLS: global longitudinal strain; HFpEF: heart failure with preserved ejection fraction; HFrEF: heart failure with reduced ejection fraction; HR: heart rate; LV: left ventricular; LVEF: left ventricular ejection fraction; LVOT VTI: left ventricular outflow tract velocity time integral; MAP: mean arterial pressure; MBF: myocardial blood flow; MEE: myocardial external efficiency; mPAP: mean pulmonary arterial pressure; MVO2: myocardial oxygen consumption; PH: pulmonary hypertension; PCWP: pulmonary capillary wedged pressure; PH: pulmonary hypertension; PrSO2: peripheral regional tissue oxygenation; PVR: pulmonary vascular resistance; RAP: right atrial pressure; RCT: randomized controlled trial; RV: right ventricle; SV: stroke volume; SVO2: mixed venous oxygen saturation; SVR: systemic vascular resistance; TAPSE: tricuspid annular peak systolic excursion; VIS: Vasoactive Inotropic Score; βHB: β-hydroxybutyrate.

**Table 7 nutrients-17-03125-t007:** Overview of studies on inflammatory disorders.

Condition	Reference (Year)	Study Design	Participants (n)	Type of Exogenous Ketone	Dose of Exogenous Ketone	Blood βHB Level	Primary Outcomes	Main Findings	Main Limitations
**COVID-19-related ARDS**	Shahtaghi el al. (2024) [82]	Single-blinded RCT (vs. placebo)	75 (intervention: 38; placebo: 37)	βHB formulation (unspecified)	25 g 2/day during 5 days	Tested +120 min post-prandial. Significant increase in mean βHB to 5.26 (+/− 0.2) mmol/L	Pro-inflammatory cytokines (IL-1β, IL-6, TNF-α, IL-18) and anti-inflammatory cytokines (IL-10) from baseline to day 5	- No significant difference in body temperature, HR and MAP - Significant increment of SpO_2_ on day 3 and increase in PaO_2_ and P/F ratio in βHB group - Significant decrease in all pro inflammatory cytokines and increase in anti-inflammatory cytokine in βHB group - Significantly reduced duration of hospitalization and muscle fatigue on the 5th day in βHB group	- Single-blinded study - Limited dose and duration of treatment

Abbreviations: ARDS: acute respiratory distress syndrome; COVID-19: coronavirus disease 2019; IL: Interleukin; MAP: mean arterial pressure; SpO2: peripheral saturation in oxygen; PaO2: arterial partial pressure of oxygen; P/F ratio: arterial oxygen partial pressure to fractional inspired oxygen ratio; RCT: randomized controlled trial; TNF: tumor necrosis factor; βHB: β-hydroxybutyrate.

## Data Availability

The raw data supporting the conclusions of this article will be made available by the authors on request.

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
