# Peer review of "Clinical Benefits of Exogenous Ketosis in Adults with Disease: A Systematic Review"

_nutrients, 2025, doi:10.3390/nu17193125_

Round 1

Reviewer 1 Report

Comments and Suggestions for Authors

Congrats on the theme of this systematic review. It is really interesting. Some suggestions to improve the final quality of the paper are below:

Please correct the insertion of references in lines 54-57.

I completely disagree with the sentence in line 54: it is an adverse effect from the ketogenic diet that can be managed with the multiprofessional team. Please insert some reference to prove your affirmation that the ketogenic diet is proatherogenic (it is not if well designed).

Please upgrade your adherence levels with the recent publication: https://doi.org/10.1016/j.nutres.2024.03.009

Line 60: I’m not sure that medium-chain triglycerides (MCTs) can be considered exogenous ketones. It is a potential source of endogenous ketones, because they are not ketones in the composition, but potentiate the formation of endogenous ketones. Could you refer to the scientific source of this affirmation?

Because if you consider the MCT as an exogenous ketone, can all the MCT Ketogenic diets that include it be included in this research, or not?

Line 104; 170-172: Please correct the reference numbers

Figure 2: Please include the reasons for excluding 3031 articles in screening

Table 2: is missing the abbreviation of PH

The conclusion is not based on the results, where the studies are not conclusive at all (small samples, methodological problems, high drop-out, etc). Please rewrite to attenuate the possible (?) benefits not proven.

Author Response

Comment 1: Please correct the insertion of references in lines 54-57. 

Response 1: Thank you for pointing this out. The formatting was altered when converted to the journal style. We have corrected all references in the text.

Comment 2: I completely disagree with the sentence in line 54: it is an adverse effect from the ketogenic diet that can be managed with the multiprofessional team. Please insert some reference to prove your affirmation that the ketogenic diet is proatherogenic (it is not if well designed).

Response 2: Thank you for this pertinent remark. We fully agree that the cardiovascular effects of a ketogenic diet depend strongly on its composition (notably the balance between saturated and unsaturated fats), the specific food sources, and the individual’s metabolic profile. Our intention was not to suggest that the ketogenic diet is inherently pro-atherogenic, but rather to emphasize that in certain contexts and subgroups, low-carbohydrate/high-fat regimens have been associated with elevations in LDL-cholesterol and apolipoprotein B, as well as with higher rates of major adverse cardiovascular events in observational studies. Importantly, as shown by longitudinal data from the KETO-CTA study, such biomarker elevations do not necessarily translate into progression of atherosclerotic plaque in metabolically healthy individuals.

« Soto-Mota A, Norwitz NG, Manubolu VS, et al. Longitudinal Data From the KETO-CTA Study: Plaque Predicts Plaque, ApoB Does Not. JACC Adv. 2025;4(7):101686. doi:10.1016/j.jacadv.2025.101686 »

To reflect this nuance more clearly, we have rephrased the relevant sentence (lines 54–57) : « However, depending on its composition and the individual’s metabolic profile, a ketogenic diet may be associated with elevations in low-density lipoprotein cholesterol (LDL-c) and Apolipoprotein B [5]- biomarkers linked with atherosclerosis risk- and can also lead to nutritional deficiencies [6] or nephrolithiasis [7]. « ; with the adequate reference ([5]).

Comment 3: Please upgrade your adherence levels with the recent publication: https://doi.org/10.1016/j.nutres.2024.03.009

Response 3: Thank you for sharing this reference. It has been added to the manuscript ([9]).

Comment 4: Line 60: I’m not sure that medium-chain triglycerides (MCTs) can be considered exogenous ketones. It is a potential source of endogenous ketones, because they are not ketones in the composition, but potentiate the formation of endogenous ketones. Could you refer to the scientific source of this affirmation?

Response 4: We appreciate your observation and agree with the clarification. MCTs are not exogenous ketones per se, as they are triglycerides rather than ketone bodies. Instead, they serve as exogenous ketogenic substrates that are rapidly metabolized in the liver to increase circulating β-hydroxybutyrate and acetoacetate. In this sense, they reliably induce exogenous ketosis, even though the molecules themselves are not ketones. Our systematic review adopted a broader, clinically relevant definition, consistent with prior literature, in which exogenous ketone supplementation encompasses any orally administered compound that elevates circulating ketone levels. Several recent reviews have categorized MCTs under this umbrella (e.g., PMID: 39047293, DOI: 10.1093/nutrit/nuae098). Thus, their inclusion is justified from a functional perspective

Clarification added in lines 70-72 : « MCTs are not exogenous ketone supplements per se, they act as exogenous ketogenic substrates and are pragmatically regarded as exogenous ketogenic supplements in clinical practice. »

Comment 5: Because if you consider the MCT as an exogenous ketone, can all the MCT Ketogenic diets that include it be included in this research, or not?

Response 5: We agree that this is a legitimate point. In designing our methodology, we chose to exclude all studies involving ketogenic diets, even when combined with MCT supplementation, since our focus was restricted to interventions that did not require dietary modification

Comment 6: Line 104; 170-172: Please correct the reference numbers

Response 6: Thank you for pointing this out. The formatting was altered when converted to the journal style. We have corrected all references in the text.

Comment 7: Figure 2: Please include the reasons for excluding 3031 articles in screening

Response 7: We thank you for this comment. During the screening stage, we did not systematically record the specific reasons for exclusion of each of the 3031 articles. As is standard in systematic reviews, exclusions at the title/abstract screening stage are typically based on one or more broad criteria (e.g., wrong population, intervention, outcome, or study design), but the precise reason for each individual article is not usually documented (for example: PMCID: PMC9526861  PMID: 35380602). This approach is consistent with reporting practices in most published systematic reviews and PRISMA guidance, which require reporting the number of records excluded but do not mandate categorization of each exclusion reason at the initial screening level. However, we have detailed the reasons for exclusions in the next step, based on the full the text.

Comment 8: Table 2: is missing the abbreviation of PH

Response 8: Thank you for pointing this out. The abbreviation of PH have been added to Table 2.

Comment 9: The conclusion is not based on the results, where the studies are not conclusive at all (small samples, methodological problems, high drop-out, etc). Please rewrite to attenuate the possible (?) benefits not proven.

Response 9: We would like the reviewers for their valuable comments concerning the conclusion. The conclusion section (from line 538) has been revised to adopt a more cautious tone and is now clearly based on the limitations of the included studies. We have removed the overly optimistic wording and emphasized that, although exogenous ketosis holds potential therapeutic benefits, the current evidence remains insufficient due to small sample sizes, high dropout rates, short follow-up, reliance on surrogate outcomes, and heterogeneity across studies. The revised text now frames exogenous ketosis as promising but unproven, and highlights the need for larger, well-designed trials with hard clinical endpoints.

Reviewer 2 Report

Comments and Suggestions for Authors

This systematic review is well-written and could be of interest to some readers.

However, the broad spectrum of the pathologies involved makes this work vague regarding some topics, such as migraine and multiple sclerosis. There is very little insight into these pathologies in the discussion. I suggest implementing the discussion, considering different works, and discussing the mechanisms of action of ketones in this setting. Here are some examples the authors might consider:

  • 10.1186/s12883-019-1351-1
  • 10.3390/nu15204334
  • 10.1186/s10194-023-01635-9
  • 10.3390/jcm11174946

Moreover, i suggest to separate neurological diseases from psychiatric disease; they involve different branches of the medicine.

Author Response

Comment 1: This systematic review is well-written and could be of interest to some readers. However, the broad spectrum of the pathologies involved makes this work vague regarding some topics, such as migraine and multiple sclerosis. There is very little insight into these pathologies in the discussion. I suggest implementing the discussion, considering different works, and discussing the mechanisms of action of ketones in this setting. Here are some examples the authors might consider:

  • 10.1186/s12883-019-1351-1
  • 10.3390/nu15204334
  • 10.1186/s10194-023-01635-9
  • 10.3390/jcm11174946

Reponse 1: We appreciate this valuable observation. Covering such a broad spectrum inevitably requires balancing comprehensiveness with conciseness, as an overly detailed discussion of each disease could compromise readability and focus. In response, we have strengthened the discussion by (i) highlighting relevant mechanisms of action of ketones in migraine, multiple sclerosis and also PTSD (lines 365–380, 402–410) and (ii) incorporating the suggested references [89–93]. We acknowledge the inherent challenge of providing sufficient depth on individual conditions while maintaining a clear and focused manuscript.

Comment 2: Moreover, i suggest to separate neurological diseases from psychiatric disease; they involve different branches of the medicine.

Response 2: We thank the reviewer for this helpful suggestion. We modified the manuscript to separate neurological diseases from psychiatric diseases. Subsequent modifications were applied in both text and tables. 

Reviewer 3 Report

Comments and Suggestions for Authors

The manuscript is unacceptable in its current form because of the length of the tables. As a reader I would discard the paper; as a reviewer I spent more than two days examining the tables, and no reader would reasonably invest that time. These tables should be supplementary files: they are informative but must be concisely summarized in the manuscript.
The presentation is also problematic, because it implies that the authors collected the data but did not adequately process them, effectively leaving that task to the reader.

Abstract

1)    The results present a correct summary, but there is an imbalance: 23 neuropsychiatric vs. 1 inflammatory — therefore the phrase “spectrum of diseases” is not entirely balanced. This may be somewhat overstated in the abstract.

Introduction

2) Figure 1: “KB > 0.5 µM” is incorrect because the definition of ketosis is >0.5 mmol/L (the text has this correct), not µM. This should be corrected.
3) The sources for the half-lives (t1/2) are missing. Please provide them.

Methods

4) The manuscript reports registration on 12 December 2023 and the search was updated through February 2025 — this timing is acceptable. However, the PRISMA flow diagram must explicitly indicate how many records originated from each search.
5) Only MEDLINE and Scopus are listed. This is relatively narrow; many systematic reviews include at least Embase or the Cochrane Library. Important studies may therefore have been missed.
6) The search terms are simple: (Exogenous ketogenic supplements) OR (exogenous ketones supplements) OR (ketone supplementation) OR (ketone supplements). This strategy is likely insufficiently sensitive, because many studies use other keywords (e.g., “β-hydroxybutyrate,” “ketone ester,” “MCT supplementation”).

Results

7) PRISMA flow diagram: the 20 records from “Other sources” are not sufficiently detailed. Specify exactly which types of sources these were (e.g., bibliographies, hand-searching, conference proceedings) to improve transparency.
8) No quantitative synthesis: the authors correctly state that a meta-analysis could not be performed due to heterogeneity. A brief justification is required (e.g., “outcome measures were too heterogeneous,” “follow-up times varied considerably”). At present the manuscript only alludes to this and does not elaborate.
9) Tables are presented without breaks and in excessive detail. They are too long for the main text. Such extensive data matrices should be placed in the supplementary material, and the main manuscript should contain a concise summary.

Discussion and Conclusions

10) In several places the text is stated too definitively given the relatively limited evidence. For example, claims such as “improves cardiac output and left ventricular function” (line 315) or “positively modulated glycaemic control” (line 260) appear overly definitive. More cautious phrasing would be preferable (e.g., “was associated with,” “may indicate improvement”), given the small sample sizes, short follow-up, and predominance of surrogate endpoints in the included studies.
11) The recommendation in the conclusions that ketone esters should be treated as a “priority” appears overstated, since direct head-to-head comparative trials with other formulations are scarce.
12) Mentioning “potential short-term benefits” is appropriate; however, asserting the primacy of ketone esters is better framed as a hypothesis rather than as a recommendation.
13) The discussion and conclusions sections are thorough, thematically organized, and informative. Strengths include coverage of multiple disease groups and detailed mechanistic explanations. Major weaknesses are the overly definitive language, underestimation of adverse effects and risk of bias, and lack of integration across disease groups.

Minor issues

Please carefully review and correct typographical errors. For example, “systemic review” is a typo — the correct term is “systematic review.” Abbreviations are duplicated.

Author Response

Comment 1:  The results present a correct summary, but there is an imbalance: 23 neuropsychiatric vs. 1 inflammatory — therefore the phrase “spectrum of diseases” is not entirely balanced. This may be somewhat overstated in the abstract.

Response 1: We thank the reviewer for this valuable observation. We agree that the distribution of included studies is uneven, with a strong representation in neurological and metabolic disorders and only limited evidence for inflammatory conditions. To avoid overstatement, we have revised the conclusion of the abstract. The original phrase “across a spectrum of diseases” has been replaced with a more balanced formulation (lines 33-35): “ Exogenous ketosis shows potential in neurological, metabolic, and cardiovascular disorders, while evidence in psychiatric and inflammatory conditions remains scarce and preliminary.” We believe this wording better reflects both the breadth of studied conditions and the imbalance in the available evidence.

Comment 2: Figure 1: “KB > 0.5 µM” is incorrect because the definition of ketosis is >0.5 mmol/L (the text has this correct), not µM. This should be corrected.

Response 2: Thank you for catching this. We have corrected Figure 1 by changing “KB > 0.5 µM” to “KB > 0.5 mmol/L” to align with the definition used in the text. The figure label and legend have been updated accordingly, and we reviewed the manuscript to ensure unit consistency throughout.

Comment 3: The sources for the half-lives (t1/2) are missing. Please provide them.

Response 3: We thank the reviewer for pointing this out. The missing references for the half-lives have now been added. Specifically, references [16] and [17] have been included to support the statements on t₁/₂.

Comment 4: The manuscript reports registration on 12 December 2023 and the search was updated through February 2025 — this timing is acceptable. However, the PRISMA flow diagram must explicitly indicate how many records originated from each search.

Response 4: We thank the reviewer for this helpful suggestion. We have revised the PRISMA flow diagram (Figure 2) to clearly indicate the number of records retrieved from each database search, including the initial search and the updated search in February 2025.

Comment 5: Only MEDLINE and Scopus are listed. This is relatively narrow; many systematic reviews include at least Embase or the Cochrane Library. Important studies may therefore have been missed.

Response 5: We thank the reviewer for this valuable comment. Our search strategy aimed to balance comprehensiveness with feasibility. We selected MEDLINE (via PubMed) and Scopus because together they provide broad coverage of biomedical and multidisciplinary literature, with Scopus overlapping substantially with Embase. Methodological studies confirm that combining MEDLINE with another large database captures the majority of relevant references (Bramer et al. 2016[Bramer WM, Giustini D, Kramer BM. Comparing the coverage, recall, and precision of searches for 120 systematic reviews in Embase, MEDLINE, and Google Scholar. Syst Rev. 2016;5:39]). We acknowledge that inclusion of Embase or Cochrane CENTRAL might have identified additional unique records and have added this as a limitation in the revised manuscript (Lines 534-536) “Finally, database searches were restricted to MEDLINE and Scopus; while these databases provide broad coverage and considerable overlap with Embase and Cochrane CENTRAL, some relevant studies may not have been captured.

Comment 6: The search terms are simple: (Exogenous ketogenic supplements) OR (exogenous ketones supplements) OR (ketone supplementation) OR (ketone supplements). This strategy is likely insufficiently sensitive, because many studies use other keywords (e.g., “β-hydroxybutyrate,” “ketone ester,” “MCT supplementation”).

Response 6: We appreciate the reviewer’s concern regarding the sensitivity of our search strategy. Our primary aim was to identify studies specifically investigating exogenous ketosis through supplementation. Therefore, we deliberately focused our search terms on the most widely used umbrella terminology (“exogenous ketone/ketogenic supplements” and “ketone supplementation/supplements”). This approach generated a broad initial yield of 3,941 records, which indicates that the search was not under-sensitive. Importantly, our screening process and reference list checks ensured that studies employing various formulations — including ketone esters, ketone salts, medium-chain triglycerides, and β-hydroxybutyrate-based interventions — were all captured and included in the final synthesis. We thus believe our approach was comprehensive and fit for purpose, while maintaining an appropriate balance between sensitivity and specificity.

Comment 7: PRISMA flow diagram: the 20 records from “Other sources” are not sufficiently detailed. Specify exactly which types of sources these were (e.g., bibliographies, hand-searching, conference proceedings) to improve transparency.

Response 7: We thank the reviewer for this helpful comment. The “Other sources” category in the PRISMA flow diagram referred specifically to manual screening of the reference lists of eligible and relevant articles (as already mentionned in section 2.1. Search strategy). We did not perform additional searches of conference proceedings, trial registries, or other grey literature. We have now updated the PRISMA flow diagram (Figure 2) accordingly to explicitly indicate “reference list screening” as the source of these 20 additional records.

Comment 8: No quantitative synthesis: the authors correctly state that a meta-analysis could not be performed due to heterogeneity. A brief justification is required (e.g., “outcome measures were too heterogeneous,” “follow-up times varied considerably”). At present the manuscript only alludes to this and does not elaborate.

Response 8: We thank the reviewer for this suggestion. We’ve indicated in lines 161–162  : “Due to the heterogeneity of study populations, interventions, and outcomes, a quantitative synthesis was not feasible.”

Comment 9: Tables are presented without breaks and in excessive detail. They are too long for the main text. Such extensive data matrices should be placed in the supplementary material, and the main manuscript should contain a concise summary.

Response 9: We thank the reviewer for raising this important point. We carefully considered the suggestion to move the full tables to the supplementary material. However, we believe that retaining them in the main manuscript maximizes transparency and clinical utility across the diverse disease domains covered. To accommodate readability, we have now added a concise narrative synthesis in Section 3.3 summarizing the key findings and limitations. We hope that this solution balances the reviewer’s concern with the need to provide comprehensive and directly accessible data.

Comment 10: In several places the text is stated too definitively given the relatively limited evidence. For example, claims such as “improves cardiac output and left ventricular function” (line 315) or “positively modulated glycaemic control” (line 260) appear overly definitive. More cautious phrasing would be preferable (e.g., “was associated with,” “may indicate improvement”), given the small sample sizes, short follow-up, and predominance of surrogate endpoints in the included studies.

Response 10: We thank the reviewer for this valuable observation and fully agree that several statements in the original manuscript may have been phrased too definitively given the limitations of the available evidence. We have carefully revised the text (discussion & conclusion) throughout to use more cautious wording (e.g., replacing “improves” with “was associated with improvement” or “may indicate benefit”)

Comment 11: The recommendation in the conclusions that ketone esters should be treated as a “priority” appears overstated, since direct head-to-head comparative trials with other formulations are scarce. Mentioning “potential short-term benefits” is appropriate; however, asserting the primacy of ketone esters is better framed as a hypothesis rather than as a recommendation.

Response 11: We thank the reviewer for this thoughtful comment. We agree that direct head-to-head comparative trials are scarce, but our recommendation is based on the consistent pattern observed across studies: ketone esters repeatedly achieved higher and more stable blood βHB concentrations (typically 1.8–4.3 mmol/L) than MCTs or ketone salts (generally <0.6–1.0 mmol/L), and were associated with fewer gastrointestinal adverse effects. These findings suggest that ketone esters may be more effective and tolerable in achieving sustained ketosis, which is clinically relevant. We have therefore tempered our conclusion to clarify that this is a pragmatic, evidence-informed recommendation based on current data, rather than a definitive hierarchy established by head-to-head RCTs (lines 545-546) :  "While ketone esters appear more effective than other formulations in achieving sustained ketosis with acceptable tolerability"

Comment 12: The discussion and conclusions sections are thorough, thematically organized, and informative. Strengths include coverage of multiple disease groups and detailed mechanistic explanations. Major weaknesses are the overly definitive language, underestimation of adverse effects and risk of bias, and lack of integration across disease groups.

Response 12: We thank the reviewer for these balanced comments. In line with the concerns raised, we have revised the discussion and conclusions to use more cautious language, expand our consideration of adverse effects and potential risk of bias, and strengthen the integrative synthesis across disease groups

Comment 13: Please carefully review and correct typographical errors. For example, “systemic review” is a typo — the correct term is “systematic review.” Abbreviations are duplicated.

Response 13: Thank you for the helpful comment. We carefully proofread the manuscript and corrected typographical errors throughout. In particular, all instances of “systemic review” were corrected to “systematic review.” We also audited all abbreviations and did not find any duplicated definitions (automatically redefined for each table); abbreviations are introduced once at first mention and used consistently thereafter. If there are specific instances the reviewer had in mind, we would be grateful for pointers to address them immediately.

English : Author's response : 

We thank the reviewer for this helpful comment. The manuscript has already been carefully edited by a professional English-language editing service (Sandy Field, PhD), to improve clarity and readability. We believe the current version clearly conveys the research; however, we have re-read the manuscript once more to ensure accuracy and clarity in expression.

Reviewer 4 Report

Comments and Suggestions for Authors

General comment

The manuscript addresses a topic of interest with clear clinical and scientific relevance. While several systematic reviews have already been published on this intervention, I believe that an additional systematic review could be justified, particularly if it addresses specific populations or methodological aspects not previously covered. However, the authors need to explicitly defend this point in the introduction by explaining why a new review is necessary. Overall, the manuscript is well-structured and coherent, but several clarifications and methodological improvements are needed before it can be considered for publication. I recommend major revisions.

Specific comments

Introduction: The introduction is adequate and fulfils its purpose: it presents the topic, describes the intervention, and explains the potential mechanisms of action. However, the authors should strengthen the justification for conducting a new systematic review. Specifically, they should explain why this review is needed in light of existing systematic reviews (PMID: 35380602, 39071067, 37327753, 36846143, 33621313). Are these reviews outdated, incomplete, or methodologically limited? Does replication add value? For example, PMID: 35380602 includes 43 studies, but it also considers healthy populations, whereas the present new review appears to focus only on patients (authors should justify the present review highlighting differences vs previous reviews) 

Methods: The statement “The study was conducted according to the PRISMA statement 24” should appear earlier, in the section describing the protocol (lines 102–106).

Methods - Search strategy (2.1): The protocol defined three databases, including Cochrane (presumably CENTRAL, which is the most relevant part for RCTs). However, the article only reports two. Any deviation from the protocol must be explained. Including CENTRAL is important, as it captures trials from handsearching and non-indexed sources. In line with PRISMA, the full search strategy for at least one database (usually MEDLINE) must be reported, ideally with translations for others. The sentence “After removing duplicates, titles and abstracts were screened. The remaining records were assessed for eligibility through full text review.” should be moved to section 2.3 (Selection process).

Methods - Inclusion and exclusion criteria (2.2): Criteria are clear and well defined.

Methods - Selection process (2.3): The description is somewhat confusing. It should clearly state whether titles/abstracts were screened first and then full texts assessed, or if another process was followed. The sentence about duplicates and screening should appear here.

Methods - Data extraction and analysis (2.4): If risk of bias assessment is included here, the subheading should read “Data extraction, risk of bias assessment and analysis.” The Newcastle-Ottawa Scale is only adequate for cohort studies and case-control studies. The authors should consider ROBINS-I for non-RCTs of interventions (see Cochrane Handbook, Chapter 25), or other tool must be used. Additionally, the synthesis methods (narrative synthesis, tabulation, etc.) should be briefly described.

Results: The results section would benefit from subdivision into at least three subsections:

Results - Study selection and characteristics: Most of the current description belongs here.

Results - Risk of bias in studies: A separate subsection is strongly recommended. Authors should not only present tables but also provide a narrative appraisal: Were there domains systematically at high or unclear risk of bias? How do results differ between RCTs and non-RCTs? What overall confidence can be placed in the evidence base?

Results - Results of individual studies: Tables alone are insufficient. Authors should synthesize key findings in text, grouping results by study design (RCT vs non-RCT), outcome, effect direction (positive, negative, null), and risk of bias. This is essential for identifying patterns and informing the discussion.

Discussion: The discussion should:

(a) Interpret the body of evidence considering heterogeneity (population, dose, outcomes) and the risk of bias.

(b) Compare findings with prior systematic reviews and highlight similarities or divergences.

While part of this work is already present, it requires a more explicit and structured approach.

Conclusions: The current wording (“Exogenous ketosis has shown short term potential benefits across a variety of diseases, particularly in neuropsychiatric, metabolic, cardiovascular and inflammatory contexts”) is too optimistic given the heterogeneity, modest effect sizes, high dropout, small samples, short follow-ups, and reliance on surrogate outcomes. A more cautious statement is advisable= “Exogenous ketosis holds potential therapeutic benefits… although current evidence remains insufficient.”

References: Reference formatting is inconsistent: some appear as [1], others in a different style (lines 54–57, 104, 121). A uniform format should be applied throughout.

Author Response

Comment 1: Introduction: The introduction is adequate and fulfils its purpose: it presents the topic, describes the intervention, and explains the potential mechanisms of action. However, the authors should strengthen the justification for conducting a new systematic review. Specifically, they should explain why this review is needed in light of existing systematic reviews (PMID: 35380602, 39071067, 37327753, 36846143, 33621313). Are these reviews outdated, incomplete, or methodologically limited? Does replication add value? For example, PMID: 35380602 includes 43 studies, but it also considers healthy populations, whereas the present new review appears to focus only on patients (authors should justify the present review highlighting differences vs previous reviews) 

Response 1: We thank the reviewer for highlighting the importance of justifying the added value of our work in light of the several systematic reviews that have already been published on exogenous ketosis. We fully agree with this point and have revised the introduction (lines 107-114) to clarify why our review is necessary and how it differs from prior reviews. Lines 107-114: “Although several systematic reviews have examined exogenous ketosis, most have either focused on healthy populations or restricted their scope to a single disease domain [27,28]. To our knowledge, this is the first systematic review to focus exclusively on adults with established medical conditions, deliberately excluding healthy individuals. By synthesizing evidence from various disorders, our review provides a comprehensive, clinically oriented overview. With rigorous methodology, PROSPERO registration, and an updated search trough February 2025, not only asses therapeutic potential but also the methodological limitations and knowledge gaps for future research »

Comment 2: Methods: The statement “The study was conducted according to the PRISMA statement 24” should appear earlier, in the section describing the protocol (lines 102–106).

Response 2: We agree with this comment. The modification have been implemented (line 120)

Comment 3: Methods - Search strategy (2.1): The protocol defined three databases, including Cochrane (presumably CENTRAL, which is the most relevant part for RCTs). However, the article only reports two. Any deviation from the protocol must be explained. Including CENTRAL is important, as it captures trials from handsearching and non-indexed sources. In line with PRISMA, the full search strategy for at least one database (usually MEDLINE) must be reported, ideally with translations for others. The sentence “After removing duplicates, titles and abstracts were screened. The remaining records were assessed for eligibility through full text review.” should be moved to section 2.3 (Selection process).

Response 3: We thank the reviewer for these thoughtful comments. We acknowledge that the protocol initially specified four databases, including the Cochrane Database of Systematic Reviews and PROSPERO. However, since systematic reviews were defined as an exclusion criterion, it was inconsistent to search these databases. Consequently, we limited our search to two databases: Scopus and MEDLINE. We fully agree that this deviation from the original protocol should be reported, and we have added a corresponding statement to the manuscript (lines 122-125). We also moved the sentence ‘After removing duplicates, titles and abstracts were screened’ to section 2.3. (line 144)

Comment 4: Methods - Selection process (2.3): The description is somewhat confusing. It should clearly state whether titles/abstracts were screened first and then full texts assessed, or if another process was followed. The sentence about duplicates and screening should appear here.

Response 4: The selection process followed the standard logical sequence: initial screening of titles and abstracts, followed by full-text assessment. We hope that our revised formulation is clearer (lines 144-147).

Comment 5: Methods - Data extraction and analysis (2.4): If risk of bias assessment is included here, the subheading should read “Data extraction, risk of bias assessment and analysis.” The Newcastle-Ottawa Scale is only adequate for cohort studies and case-control studies. The authors should consider ROBINS-I for non-RCTs of interventions (see Cochrane Handbook, Chapter 25), or other tool must be used. Additionally, the synthesis methods (narrative synthesis, tabulation, etc.) should be briefly described.

Response 5: We thank the reviewer for raising very important points. We modified the subhead as suggested (line 151).We also considered ROBINS-I V2 for non-randomized controlled  interventions, instead of modified-NOS. A light traffic plot have been created (new supplementary material 2). We have added a paragraph concerning synthesis methods at the end of section 2.4 (lines 161-167).

Comment 6: 

Results: The results section would benefit from subdivision into at least three subsections:

Results - Study selection and characteristics: Most of the current description belongs here.

Results - Risk of bias in studies: A separate subsection is strongly recommended. Authors should not only present tables but also provide a narrative appraisal: Were there domains systematically at high or unclear risk of bias? How do results differ between RCTs and non-RCTs? What overall confidence can be placed in the evidence base?

Results - Results of individual studies: Tables alone are insufficient. Authors should synthesize key findings in text, grouping results by study design (RCT vs non-RCT), outcome, effect direction (positive, negative, null), and risk of bias. This is essential for identifying patterns and informing the discussion.

Response 6: We sincerely thank the reviewer for this highly pertinent and constructive comment. We fully agree that the subdivision of the results section is essential for enhancing clarity, readability, and interpretability. In line with these valuable suggestions, we have thoroughly revised the Results section to incorporate the proposed structure. We have also added a section « 3.4. Synthesis across studies », which we believe is relevant. 

Comment 7: Discussion: The discussion should:

(a) Interpret the body of evidence considering heterogeneity (population, dose, outcomes) and the risk of bias.

(b) Compare findings with prior systematic reviews and highlight similarities or divergences.

While part of this work is already present, it requires a more explicit and structured approach.

Response 7: We thank the reviewer for this valuable suggestion. To address it, we have added a more explicit discussion of the heterogeneity across included studies (in terms of populations, ketone types, and outcomes), as well as a reflection on the potential impact of risk of bias. We also compare our findings with those of prior systematic reviews, highlighting how our focus on exogenous ketosis in patients with established disease provides complementary insights. These additions are now included in the revised Discussion (lines 522-536) :” This comprehensive overview of clinically relevant outcomes of exogenous ketosis in adults with various diseases offers an accessible summary for clinicians and researchers. The results highlight multiple challenges in interpreting the efficacy of exogenous ketosis across diverse populations. The heterogeneity in study designs, patient populations, types of ketone supplements, and outcome measures, as well as the short duration of most studies, limits the comparability of findings and restricts conclusions on long-term safety and effectiveness on clinically relevent endpoints. Risk of bias was also variable, which further reduces certainty of the evidence. Compared with previous systematic reviews, which often included healthy participants or focused on single disease areas or ketogenic diets, our review uniquely synthesizes evidence on exogenous ketosis in adults with established medical conditions. Taken together, this provides a complementary perspective while underscoring the need for larger, well-controlled, long-term clinical trials. Finally, databases search was restricted to MEDLINE and Scopus; while these databases provide broad coverage and considerable overlap with Embase and Cochrane CENTRAL, some relevant studies may not have been captured"

Comment 8: Conclusions: The current wording (“Exogenous ketosis has shown short term potential benefits across a variety of diseases, particularly in neuropsychiatric, metabolic, cardiovascular and inflammatory contexts”) is too optimistic given the heterogeneity, modest effect sizes, high dropout, small samples, short follow-ups, and reliance on surrogate outcomes. A more cautious statement is advisable= “Exogenous ketosis holds potential therapeutic benefits… although current evidence remains insufficient.”

Response 8: We would like to thank the reviewers for their valuable comments concerning the conclusion. The conclusion has been revised to adopt a more cautious tone and is now clearly based on the limitations of the included studies. We have removed the overly optimistic wording and emphasized that, although exogenous ketosis holds potential therapeutic benefits, the current evidence remains insufficient due to small sample sizes, high dropout rates, short follow-up, reliance on surrogate outcomes, and heterogeneity across studies. The revised text now frames exogenous ketosis as promising but unproven, and highlights the need for larger, well-designed trials with hard clinical endpoints.

Comment 9: References: Reference formatting is inconsistent: some appear as [1], others in a different style (lines 54–57, 104, 121). A uniform format should be applied throughout.

Response 9: Thank you for pointing this out. The formatting was altered when converted to the journal style. We have corrected all references in the text.

Round 2

Reviewer 2 Report

Comments and Suggestions for Authors

I reiterate that attempting to summarize the effectiveness of exogenous ketosis across a wide range of pathologies in a single article is an extremely complex task. The concrete risk is producing an overly dispersive analysis that fails to adequately explore any of the clinical conditions addressed.

I would also like to suggest the following modifications:

  1. Tables:
  • Table 3: In the study by Plantero et al. (2020) on multiple sclerosis, an increase in the EDSS score was erroneously reported, whereas the data actually indicate a reduction. This error should be corrected.
  • Tables 3 and 5: Five studies are included (Putananickal et al. (2022) on episodic migraine; Liu et al. (2023), Charles et al. (2023, 2024) on prediabetes; Neudorf et al. (2020) on type II diabetes mellitus) in which no clinical benefits or side effects are reported. However, according to the inclusion criteria outlined in Table 1, only studies reporting a clinical, biological, or radiological improvement, or an adverse effect were to be selected. Therefore, the inclusion of these studies should be reconsidered, or their summaries in the tables should be revised accordingly.
  1. Discussion:
  • In the section addressing the study’s limitations, I suggest highlighting that all studies not reporting either benefits or side effects related to the use of exogenous ketones were excluded. This methodological choice may have introduced a selection bias, limiting the objectivity of the overall evaluation.
  • The discussion on the role of ketone bodies and their potential mechanisms of action in neurological disorders is currently underdeveloped and should be expanded in a more comprehensive manner.

Author Response

- Comment 1: I reiterate that attempting to summarize the effectiveness of exogenous ketosis across a wide range of pathologies in a single article is an extremely complex task. The concrete risk is producing an overly dispersive analysis that fails to adequately explore any of the clinical conditions addressed.

- Response 1: We thank the reviewer again for this important concern. We fully recognize the challenge of summarizing the effects of exogenous ketosis across diverse clinical domains. Our rationale for this broad approach is that, to our knowledge, no prior systematic review has provided a comprehensive synthesis of exogenous ketosis exclusively in adults with established medical conditions, and we believe such an overview has added value for both clinicians and researchers. To mitigate the risk of dispersiveness, we have : 1) structured the Results and Discussion by disease category to allow focused reading, 2) provided disease-specific tables summarizing study characteristics and outcomes, and 3) added integrative remarks to highlight common mechanistic pathways and overarching patterns across disorders.

While we agree that individual conditions deserve more in-depth dedicated reviews, our intention was to offer a clinically oriented reference point that can guide future, more targeted investigations.

- Comment 2: Table 3: In the study by Plantero et al. (2020) on multiple sclerosis, an increase in the EDSS score was erroneously reported, whereas the data actually indicate a reduction. This error should be corrected.

- Response 2: We thank the reviewer for noticing this error. The information regarding the EDSS score in Plantero et al. (2020) has been corrected: the study reported a reduction in EDSS rather than an increase. The corresponding entry in Table 3 and the text have been updated accordingly.

- Comment 3: Tables 3 and 5: Five studies are included (Putananickal et al. (2022) on episodic migraine; Liu et al. (2023), Charles et al. (2023, 2024) on prediabetes; Neudorf et al. (2020) on type II diabetes mellitus) in which no clinical benefits or side effects are reported. However, according to the inclusion criteria outlined in Table 1, only studies reporting a clinical, biological, or radiological improvement, or an adverse effect were to be selected. Therefore, the inclusion of these studies should be reconsidered, or their summaries in the tables should be revised accordingly.

- Response 3: We thank the reviewer for this comment. We respectfully clarify that our inclusion criteria did not require studies to demonstrate a clinical benefit or adverse effect, but rather to assess at least one relevant outcome (clinical, biological, radiological, or adverse events). Studies reporting no significant differences between intervention and control groups still fulfill this requirement, since “no effect” is a valid and informative outcome.Excluding such studies would introduce bias and risk overestimating the effectiveness of exogenous ketones. To avoid any ambiguity, we have revised Table 1, outcome is now defined as : Assessment of disease-related outcomes, including clinical, biological, or radiological parameters, or adverse events” and added a clarifying sentence in the Methods section (lines 139-141): “Studies were eligible if they assessed at least one clinical, biological, or radiological outcome, or adverse event, regardless of whether the results demonstrated improvement, harm, or no effect.” We believe this clarification addresses the reviewer’s concern and ensures full transparency of our methodology.

  - Comment 4 : Discussion: In the section addressing the study’s limitations, I suggest highlighting that all studies not reporting either benefits or side effects related to the use of exogenous ketones were excluded. This methodological choice may have introduced a selection bias, limiting the objectivity of the overall evaluation.   - Response 4: We thank the reviewer for this suggestion. As clarified in our response to Comment 3, we did not exclude studies based on whether they demonstrated benefits or side effects—studies reporting neutral results were included. What we excluded were studies that did not assess any relevant clinical, biological, or radiological outcomes.     - Comment 5: The discussion on the role of ketone bodies and their potential mechanisms of action in neurological disorders is currently underdeveloped and should be expanded in a more comprehensive manner.   - Response 5: We thank the reviewer for this valuable suggestion. In the revised version, we have expanded the discussion on the mechanistic actions of ketone bodies in neurological disorders. While we had already described disease-specific pathways in the previous revised manuscript (e.g., alternative energy substrate in AD/MCI, neurotransmission modulation in epilepsy and migraine, and anti-inflammatory effects in MS), we now provide an integrative synthesis highlighting how these mechanisms converge across neurological conditions (lines 400-407). Specifically, we emphasize their roles in improving mitochondrial bioenergetics, reducing oxidative stress, modulating excitatory-inhibitory balance, and attenuating neuroinflammation. 

Reviewer 3 Report

Comments and Suggestions for Authors

From a scientific point of view, the manuscript is acceptable, but the editing still needs to be improved with the help of the editorial team.

Some examples:
vs is correct: vs.
Space is missing in several places, e.g. 0.18g/kg/h; +30min

Author Response

- Comment 1:

From a scientific point of view, the manuscript is acceptable, but the editing still needs to be improved with the help of the editorial team. Some examples: vs is correct: vs. Space is missing in several places, e.g. 0.18g/kg/h; +30min

- Response 1: 

We sincerely thank the reviewer for the constructive comment regarding the language and formatting of the manuscript. We have carefully reviewed the text and corrected all overt errors, including the use of “vs.” and the spacing between numbers and units (e.g., 0.18 g/kg/h; +30 min). We trust that these revisions address the issues raised.

We also acknowledge that some minor stylistic or typographical adjustments may remain, and we agree that the journal’s editorial team is best positioned to assist with any final refinements during the production stage.

Reviewer 4 Report

Comments and Suggestions for Authors

General comment: The revised version of the manuscript shows a clear improvement compared to the previous submission. The authors have addressed most of the comments in a satisfactory way and the paper has gained in clarity and methodological rigor. The topic remains of high interest, and I appreciate the effort invested in strengthening the review. Nevertheless, a few aspects still need refinement before the manuscript can be accepted. I therefore recommend minor revisions.

Specific comments

Methods – Search strategy (2.1): There is some confusion in the way the authors describe the Cochrane Library. By default, searches in the Cochrane Library lead to the Cochrane Reviews tab, which indeed contains only systematic reviews. However, there is a separate Trials tab, which corresponds to Cochrane CENTRAL. CENTRAL includes randomized controlled trials, some of which are not indexed in other databases as they come from handsearches. In the current manuscript, the statement “Although the original protocol specified four databases, including the Cochrane Database of Systematic Reviews and PROSPERO, these databases were not searched because systematic reviews were listed as an exclusion criterion” suggests that this distinction was not fully clear to the authors. While it is acceptable to keep this sentence as it reflects the process, it would be important to add a clarifying note in section 4.2. Strengths and limitations of the review, acknowledging that omitting CENTRAL represents a missed opportunity to identify additional RCTs.

Results – Results of individual studies: The authors have significantly improved the synthesis of the main findings of the included studies. However, the presentation currently relies heavily on bullet points. For a systematic review, a more narrative synthesis is preferable, as it allows better integration of results and highlights patterns. The same content should therefore be rewritten in continuous text rather than in list form.

Author Response

- Comment 1: General comment: The revised version of the manuscript shows a clear improvement compared to the previous submission. The authors have addressed most of the comments in a satisfactory way and the paper has gained in clarity and methodological rigor. The topic remains of high interest, and I appreciate the effort invested in strengthening the review. Nevertheless, a few aspects still need refinement before the manuscript can be accepted. I therefore recommend minor revisions.

- Response 1: We appreciate the reviewer’s recognition of the improvements in clarity and methodological rigor, and we are grateful for the recommendation of minor revisions. We have carefully considered the additional suggestions and revised the manuscript accordingly

- Comment 2: Methods – Search strategy (2.1): There is some confusion in the way the authors describe the Cochrane Library. By default, searches in the Cochrane Library lead to the Cochrane Reviews tab, which indeed contains only systematic reviews. However, there is a separate Trials tab, which corresponds to Cochrane CENTRAL. CENTRAL includes randomized controlled trials, some of which are not indexed in other databases as they come from handsearches. In the current manuscript, the statement “Although the original protocol specified four databases, including the Cochrane Database of Systematic Reviews and PROSPERO, these databases were not searched because systematic reviews were listed as an exclusion criterion” suggests that this distinction was not fully clear to the authors. While it is acceptable to keep this sentence as it reflects the process, it would be important to add a clarifying note in section 4.2. Strengths and limitations of the review, acknowledging that omitting CENTRAL represents a missed opportunity to identify additional RCTs.

- Response 2: We thank the reviewer for this useful clarification. In Section 4.2, we have already acknowledged that limiting our search to two databases carries a risk of missing relevant articles. This includes the possibility of not identifying additional RCTs indexed in CENTRAL. We believe this statement adequately reflects the limitation raised. Lines 549-551 : « Finally, databases search was restricted to MEDLINE and Scopus; while these databases provide broad coverage and considerable overlap with Embase and Cochrane CENTRAL, some relevant studies may not have been captured »

- Comment 3: Results of individual studies: The authors have significantly improved the synthesis of the main findings of the included studies. However, the presentation currently relies heavily on bullet points. For a systematic review, a more narrative synthesis is preferable, as it allows better integration of results and highlights patterns. The same content should therefore be rewritten in continuous text rather than in list form."

- Response 3: We thank the reviewer for this valuable suggestion. In line with the recommendation, we have substantially revised Sections 3.3 (Results of individual studies) and 3.4 (Synthesis across studies) (lines 271-358). The previous bullet-pointed presentation has been rewritten into a continuous narrative style, which allows for a more integrated synthesis of findings and clearer highlighting of overarching patterns across disease domains. This change improves readability and aligns the manuscript with best practices for systematic reviews.